# Decitabine cytotoxicity is promoted by dCMP deaminase DCTD and mitigated by SUMO-dependent E3 ligase TOPORS

Christopher J Carnie [ID] [1,2,6 ✉], Maximilian J Götz[3,6], Chloe S Palma-Chaundler[1,6], Pedro Weickert[3], Amy Wanders[2], Almudena Serrano-Benitez[1,2], Hao-Yi Li[3], Vipul Gupta[2], Samah W Awwad[1,2], Christian J Blum [ID] [4], Matylda Sczaniecka-Clift[2], Jacqueline Cordes[3], Guido Zagnoli-Vieira[2], Giuseppina D'Alessandro[1,2], Sean L Richards[1], Nadia Gueorguieva[1,2], Simon Lam [ID] [1,2], Petra Beli [ID] [4,5], Julian Stingele [ID] [3 ✉] & Stephen P Jackson [ID] [1,2 ✉]

## Abstract

**The nucleoside analogue decitabine (or 5-aza-dC) is used to treat several haematological cancers. Upon its triphosphorylation and incorporation into DNA, 5-aza-dC induces covalent DNA methyltransferase 1 DNA–protein crosslinks (DNMT1-DPCs), leading to DNA hypomethylation. However, 5-aza-dC's clinical outcomes vary, and relapse is common. Using genome-scale CRISPR/Cas9 screens, we map factors determining 5-aza-dC sensitivity. Unexpectedly, we find that loss of the dCMP deaminase DCTD causes 5-aza-dC resistance, suggesting that 5-aza-dUMP generation is cytotoxic. Combining results from a subsequent genetic screen in DCTD-deficient cells with the identification of the DNMT1-DPC-proximal proteome, we uncover the ubiquitin and SUMO1 E3 ligase, TOPORS, as a new DPC repair factor. TOPORS is recruited to SUMOylated DNMT1-DPCs and promotes their degradation. Our study suggests that 5-aza-dC-induced DPCs cause cytotoxicity when DPC repair is compromised, while cytotoxicity in wild-type cells arises from perturbed nucleotide metabolism, potentially laying the foundations for future identification of predictive biomarkers for decitabine treatment.**

**Keywords** Genome Stability; DNA–Protein Crosslinks; Nucleotide Metabolism; SUMO-targeted Ubiquitylation; Hypomethylating Agents
**Subject Categories** Cancer; DNA Replication, Recombination & Repair; Post-translational Modifications & Proteolysis

## Introduction

Myelodysplastic syndromes (MDS) are a heterogenous group of neoplastic disorders that represent the most common group of haematological malignancies (Bejar and Steensma, 2014). MDS is characterised by dysplasia and ineffective haematopoiesis, leading to peripheral cytopenia along with a risk of disease progression to acute myelocytic leukaemia (AML) (Arber et al, 2016). The core therapy for the management of MDS consists of the nucleoside analogues 5-azacytidine (5-aza-C) and 5-aza-deoxycytidine (5-aza-dC, also known as decitabine). Decitabine is also used for the treatment of AML and chronic myelocytic leukaemia (CML), particularly in elderly patients who are ineligible to undergo more aggressive regimens with agents such as cytarabine (cytosine arabinoside) (Fenaux and Adès, 2013; Saliba et al, 2021).

Azacytidines are generally believed to exert their therapeutic effects primarily by becoming incorporated into nascent DNA and acting as a pseudosubstrate for DNA methyltransferase 1 (DNMT1) at hemimethylated CpGs in postreplicative DNA, trapping a covalent DNA-DNMT1 reaction intermediate as a DNA–protein crosslink (DPC) (Tsujioka et al, 2015; Jüttermann et al, 1994). DPCs are highly toxic DNA lesions that interfere with chromatin-associated processes such as replication and transcription (Weickert and Stingele, 2022; Stingele et al, 2017; Carnie et al, 2024; Oka et al, 2024; van Sluis et al, 2024). Repair of these lesions requires proteolytic degradation by either the proteasome or dedicated DPC proteases of the Wss1/SPRTN family (Vaz et al, 2016; Stingele et al, 2014, 2016; Larsen et al, 2019; Reinking et al, 2020). DPC repair can be initiated in a replication-coupled manner upon DNA polymerase stalling (Stingele et al, 2014; Duxin et al, 2014; Vaz et al, 2016; Stingele et al, 2016; Lopez-Mosqueda et al, 2016). In addition, global-genome DPC repair, which repairs DNMT1-DPCs behind the replication fork upon the incorporation of 5-aza-dC into

[1]Cancer Research UK Cambridge Institute, University of Cambridge, Cambridge, UK. [2]The Gurdon Institute and Department of Biochemistry, University of Cambridge, Cambridge, UK. [3]Gene Center and Department of Biochemistry, Ludwig-Maximilians-Universität München, Munich, Germany. [4]Institute of Molecular Biology (IMB), Mainz, Germany. [5]Institute of Developmental Biology and Neurobiology (IDN), Johannes Gutenberg-Universität, Mainz, Germany. [6]These authors contributed equally: Christopher J Carnie, Maximilian J Götz, Chloe S Palma-Chaundler. ✉E-mail: chris.carnie@cruk.cam.ac.uk; stingele@genzentrum.lmu.de; steve.jackson@cruk.cam.ac.uk

nascent DNA, begins with DPC SUMOylation and subsequent ubiquitylation by the SUMO-targeted ubiquitin ligase (STUbL) RNF4, promoting DPC degradation by SPRTN and the proteasome (Borgermann et al, 2019; Sun et al, 2020; Liu et al, 2021; Weickert et al, 2023).

Degradation of crosslinked DNMT1 depletes the enzyme and thus results in DNA hypomethylation and re-expression of previously silenced tumour suppressor genes (Tsujioka et al, 2015; Daskalakis et al, 2002). 5-aza-C and 5-aza-dC are thus considered as hypomethylating agents (HMAs). Despite their widespread use, responses to HMAs vary from patient to patient for reasons that remain unclear (Treppendahl et al, 2014; Welch et al, 2016). Only around 30–50% of patients respond well to HMAs (Griffiths and Gore, 2008; Momparler et al, 1985), with a subpopulation not responding at all. This is especially problematic because HMAs are given in low doses over long treatment periods of up to 6 months before individual treatment effectiveness can be assessed (Blum, 2010); parallels can be drawn to other chemotherapeutics such as TOP1 poisons with complex mechanisms of cellular resistance that emerge throughout a treatment regimen (Zhang et al, 2022; Kumar et al, 2023). However, promising new approaches using HMAs in combination with other drugs such as the BCL2 inhibitor venetoclax are emerging (Saliba et al, 2021; DiNardo et al, 2020). Therefore, a detailed understanding of the mechanism(s) of action of HMAs and the identification of predictive biomarkers to guide individual therapy is becoming increasingly important. In addition to DNMT1-DPC induction (Cheng et al, 2018; Jüttermann et al, 1994; Šorm et al, 1964), HMAs also cause broad cytotoxicity by perturbing RNA synthesis and activating immune checkpoints (Roulois et al, 2015). 5-aza-C, a ribonucleoside, is incorporated into both RNA and DNA, the latter being dependent on reduction of 5-aza-CDP to 5-aza-dCDP by ribonucleotide reductase (Glover and Leyland-Jones, 1987; Liou et al, 2002; Van Rompay et al, 2001). The deoxyribonucleoside 5-aza-dC is phosphorylated by deoxycytidine kinase (DCK) (Stegmann et al, 1995) to 5-aza-dCMP. 5-aza-dCMP is further phosphorylated by cytidine/uridine monophosphate kinase 1 (CMPK1) and nucleoside diphosphate kinase 1/2 (NME1/2) to generate 5-aza-dCTP, which can be incorporated into nascent DNA (Momparler, 2005). Alternatively, 5-aza-dC or 5-aza-dCMP can be deaminated by dCMP deaminase (DCTD) or cytidine deaminase (CDA) to 5-aza-dUMP or 5-aza-dU, respectively (Chabot et al, 1983; Cashen et al, 2008). Interestingly, CDA is highly expressed in certain organs, such as the liver and the gut, where it deaminates HMAs (Ebrahem et al, 2012). The rapid deamination of HMAs is responsible for their short serum half-life and is believed to result in their detoxification (Ebrahem et al, 2012; Patel et al, 2021). The FDA has recently approved the combination of 5-aza-dC and the CDA inhibitor cedazurine for treating MDS, a strategy aimed at reducing the extent of 5-aza-dC deamination (Patel et al, 2021). How deamination and DPC formation precisely determine the overall mode-of-action of HMAs, however, remains unresolved.

Here, we employ a genome-wide CRISPR/Cas9 screen to map genes conferring resistance or sensitivity to decitabine treatment, uncovering a major mode of 5-aza-dC cytotoxicity that acts through its deamination by DCTD. In addition, we determine the proximal proteome of DNMT1-DPCs by isolation of proteins on nascent DNA (iPOND) and combine it with a second genome-wide genetic screen with 5-aza-dC in *DCTD* KO cells. Together, this enables us to categorise hits from our genetic screens into DPC-dependent and DPC-independent classes. Using this approach, we identify TOPORS, a SUMO1 and ubiquitin E3 ligase, as a resistance factor that is recruited to 5-aza-dC-induced DNMT1-DPCs. TOPORS recruitment is SUMO dependent but ubiquitin independent and promotes proteolysis of DNMT1-DPCs. Our findings indicate that substantial 5-aza-dC-induced cytotoxicity can arise through perturbed nucleotide metabolism rather than by DNMT1-DPC formation, but that compromised DPC repair can dramatically sensitise cells to DNMT1-DPC-mediated cytotoxicity.

# Results

## Nucleotide metabolism modulates 5-aza-dC/decitabine cytotoxicity

To profile genetic determinants of 5-aza-dC sensitivity and resistance, we performed a genome-scale CRISPR/Cas9 screen with 5-aza-dC in the CML-derived human cell line HAP1 (Fig. 1A). Based on a false discovery rate (FDR) of 0.1, our screen identified 48 genes whose individual loss conferred sensitivity to 5-aza-dC, and 11 genes whose loss conferred resistance (Fig. 1B; Dataset EV1). As expected, inactivation of *SLC29A1* or *DCK* conferred 5-aza-dC resistance, while loss of *SAMHD1* caused sensitisation (Fig. 1B). *SLC29A1* encodes ENT1, a nucleoside transporter that mediates cellular uptake of 5-aza-dC (Fig. 1C) (Saliba et al, 2021; Qin et al, 2009; Hummel-Eisenbeiss et al, 2013; Gu et al, 2021; Wu et al, 2015). DCK phosphorylates 5-aza-dC and is thus required for its subsequent incorporation into DNA (Fig. 1C). SAMHD1 is a hydrolase that cleaves dNTPs into deoxynucleosides and triphosphate and has previously been shown to hydrolyse 5-aza-dCTP (Oellerich et al, 2019) (Fig. 1C). Therefore, loss of SAMHD1 is expected to increase incorporation of 5-aza-dCTP, thereby increasing cellular sensitivity to 5-aza-dC.

Strikingly, and challenging the prevailing model of the mechanism of decitabine action, inactivation of *DCTD* or *DCTPP1* conferred 5-aza-dC resistance in our CRISPR screen (Fig. 1B,C). DCTD deaminates dCMP/5-aza-dCMP to dUMP/5-aza-dUMP (Almqvist et al, 2016), while DCTPP1 dephosphorylates dCTP/5-aza-dCTP to dCMP/5-aza-dCMP (Requena et al, 2016), thus regenerating a substrate for DCTD (Fig. 1C). To validate this, we assessed the 5-aza-dC sensitivity of *DCTD* knockout (KO) HAP1 cells in clonogenic survival assays and observed profound 5-aza-dC resistance compared to wild-type (WT) cells, while *DCK* KO cells displayed even greater resistance, as expected (Figs. 1D,E and EV1A,B). 5-aza-dUMP, generated by DCTD action on 5-aza-dCMP, thus appears to constitute a major source of 5-aza-dC-induced toxicity that is still dependent on generation of 5-aza-dCMP by DCK. 5-aza-dUMP has been demonstrated to bind, and has been suggested to inhibit, thymidylate synthetase (TYMS), which might cause increased genomic misincorporation of uracil (Almqvist et al, 2016; Requena et al, 2016), although the precise impact of the interaction between 5-aza-dUMP and TYMS remains unclear. Notably, *TYMS* inactivation also conferred strong 5-aza-dC resistance in our CRISPR screen (Fig. 1B), consistent with the idea that the interaction between 5-aza-dUMP and TYMS contributes to cytotoxicity. Together, these findings highlighted a mechanism of 5-aza-dC cytotoxicity that occurs through the generation of 5-aza-dUMP by the sequential action of DCK and DCTD.

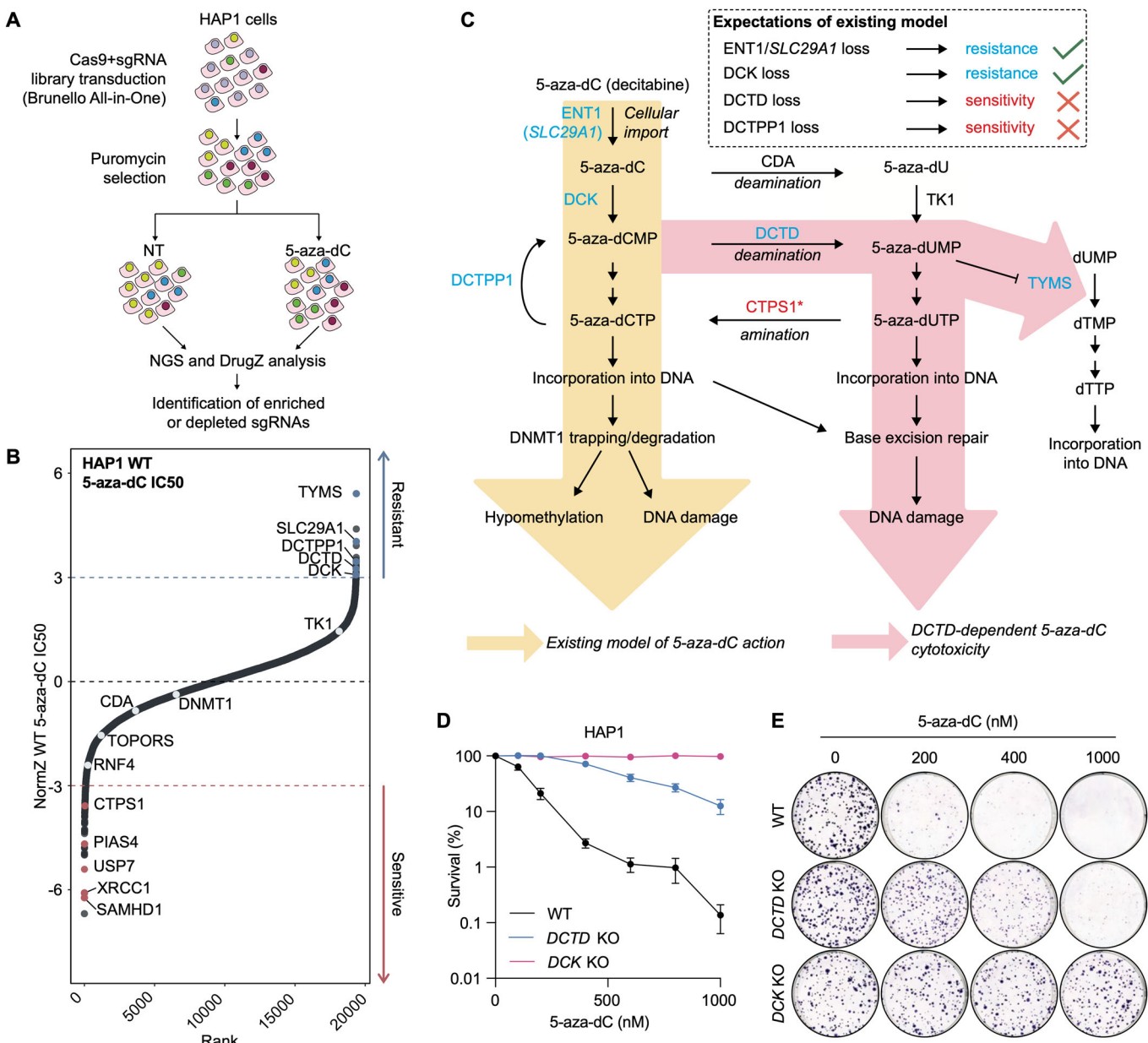

**Figure 1. Loss of DCTD confers resistance to 5-aza-dC.**

(A) Schematic outlining genome-wide CRISPR/Cas9 screen with 5-aza-dC in HAP1 cells. (B) Rank plot displaying selected hits from CRISPR/Cas9 screen with 5-aza-dC, outlined in (A); dotted lines at NormZ scores of +3/−3 indicate thresholds for resistance/sensitivity hits, respectively. (C) Schematic detailing the existing model of 5-aza-dC action and the additional action suggested by our CRISPR screen. Factors whose loss confers resistance/sensitivity to 5-aza-dC in the CRISPR screen in (B) are displayed in blue/red, respectively. * Denotes inferred enzymatic activity based on its yeast homologue. Green ticks denote alignment between our CRISPR/Cas9 screen outputs and expectations of the existing model of 5-aza-dC action, while red crosses denote disagreement between screen outputs and the existing model. (D) Clonogenic survival assays in WT, *DCTD* KO and *DCK* KO HAP1 cells treated with 5-aza-dC and stained 6 days later; n = 3 biological replicates, error bars ± SEM. (E) Representative images from (D) of cells at selected 5-aza-dC doses. Source data are available online for this figure.

## 5-aza-dC deamination drives DNMT1-independent cytotoxicity

Disruption of the *Schizosaccharomyces pombe* homologue of DCTD (Sánchez et al, 2012) has been shown to perturb normal dCTP pools, which could feasibly influence 5-aza-dCTP incorporation into DNA and subsequent DNMT1-DPC induction. However, in

our 5-aza-dC CRISPR screen, DNMT1 loss did not cause 5-aza-dC resistance (Fig. 1B), suggesting that DCTD-driven cytotoxicity is DNMT1-independent. To directly assess whether the impact of DCTD on 5-aza-dC toxicity was related to differences in DNMT1-DPC formation, we used the recently developed Purification of x-linked Proteins (PxP) assay (Weickert et al, 2023) (Fig. 2A) to monitor induction of DNMT1-DPCs by 5-aza-dC in *DCK* and

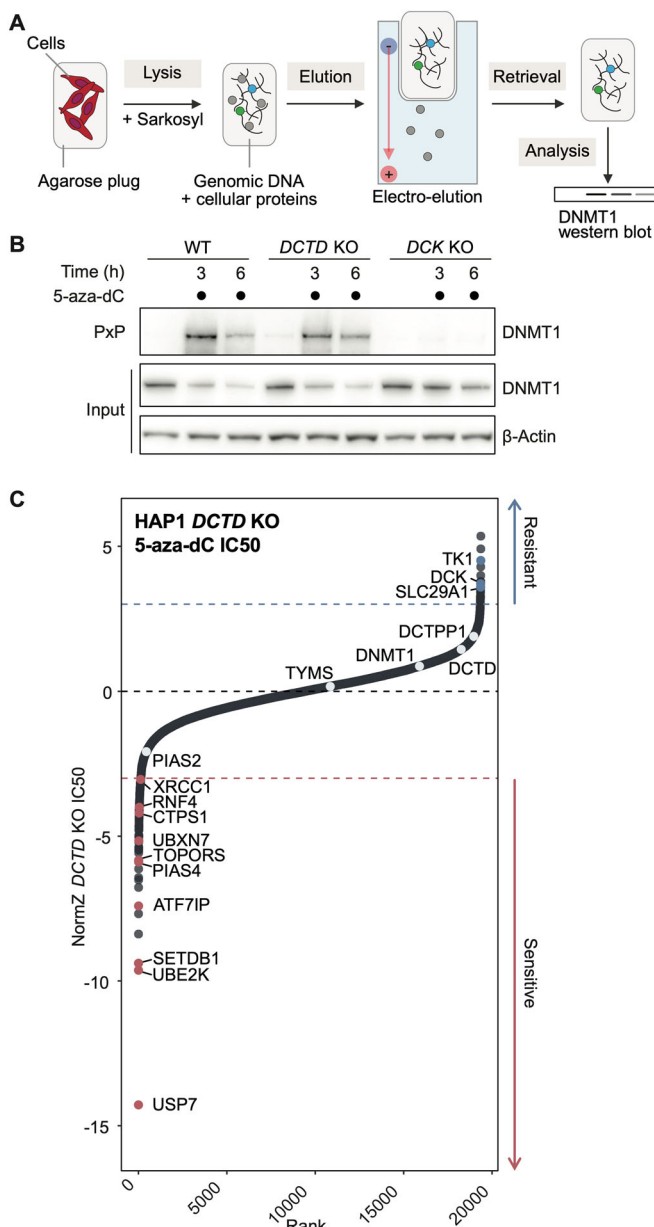

Figure 2.   5-aza-dC cytotoxicity is driven by DNMT1-dependent and
-independent mechanisms.

(A) Schematic detailing the PxP assay. Cells are harvested and cast into low-
melt agarose plugs. Plugs are transferred to the denaturing lysis buffer. After
lysis is completed, plugs are placed into the pockets of an SDS-PAGE-gel and
non-crosslinked proteins are eluted by electrophoresis. Finally, plugs are
retrieved from the gel pockets and boiled with LDS sample buffer. For DPC
detection, samples are run on SDS-PAGE gels and quantified by western
blotting. (B) DNMT1-DPC formation assessed by PxP in WT, *DCTD* KO and *DCK*
KO HAP1 cells treated with 5-aza-dC (1 µM) for the indicated times;
representative of three independent experiments. (C) Rank plot of a genome-
wide CRISPR/Cas9 screen in *DCTD* KO HAP1 cells treated with 5-aza-dC; dotted
lines at NormZ scores of +3/−3 indicate threshold for resistance/sensitivity
hits, respectively. Source data are available online for this figure.

*DCTD* KO cells. In WT cells, DNMT1-DPCs were induced within
3 h of 5-aza-dC treatment (Fig. 2B). By 6 h of treatment, the level of
DNMT1-DPCs was reduced, presumably reflecting progressive

degradation of DNMT1-DPCs. Accordingly, total cellular DNMT1
was concomitantly depleted, as evident from analysis of input
samples, reminiscent of previous observations (Weickert et al, 2023;
Patel et al, 2010) (Fig. 2B). In *DCK* KO cells however, 5-aza-dC is
not expected to be incorporated into DNA because DCK is required
for the first phosphorylation step in the generation of 5-aza-dCTP.
In agreement, we only detected minimal 5-aza-dC-induced
DNMT1-DPCs in *DCK* KO cells (Fig. 2B). It is possible that the
residual DNMT1-DPCs detected in these circumstances are caused
by 5-aza-dCTP incorporation arising downstream of deamination
of 5-aza-dC to 5-aza-dU by CDA, followed by triphosphorylation
and conversion of 5-aza-dUTP to 5-aza-dCTP by CTPS1. In line
with this idea, the *Saccharomyces cerevisiae* homologue of CTPS1
(Ctps1) has been described to convert dUTP to dCTP (Pappas et al,
1999; preprint: Guo et al, 2023) (Fig. EV1C). In contrast to our
findings with *DCK* KO cells, DNMT1-DPC induction in *DCTD* KO
cells was only slightly reduced compared with that in WT cells
(Fig. 2B), making it unlikely that the dramatic resistance of *DCTD*
KO cells to 5-aza-dC is a result of reduced DNMT1-DPC
formation. Indeed, siRNA-mediated DNMT1 depletion did not
protect WT or *DCTD* KO cells from 5-aza-dC (Fig. EV1D–F). The
minor reduction in DNMT1-DPC formation observed in *DCTD*
KO cells could be explained by differences in DNA synthesis rates
between WT and *DCTD* KO cells. Indeed, 5-ethynyl-2'-deoxyur-
idine (EdU) incorporation into nascent DNA measured by flow
cytometry revealed that *DCTD* KO cells but not WT cells show a
minor reduction in DNA synthesis upon 5-aza-dC treatment,
possibly due to stronger nucleotide imbalance resulting from
elevated dCTP levels (Diehl et al, 2022) (Fig. EV1G). As such, it
might be that *DCTD* KO cells incorporate less 5-aza-dC than
WT cells, but this does not seem to underlie the 5-aza-dC resistance
of *DCTD* KO cells, given the absence of 5-aza-dC resistance
conferred by DNMT1 depletion (Fig. EV1D–F). Together, these
data demonstrate that a substantial part of the cytotoxic action of 5-
aza-dC is DNMT1-independent and depends instead on DCTD-
mediated deamination.

DCTD has recently been shown to mediate cytotoxicity of 5'-
hydroxymethyl-deoxycytidine monophosphate (hmdCMP) through
its deamination to hmdUMP, leading to incorporation of hmdU into
DNA and subsequent DNA single-strand break (SSB) generation
during base excision repair (BER), in a manner dependent on the
glycosylase SMUG1 (Fugger et al, 2021). Notably, loss of the SSB
repair protein XRCC1 strongly sensitised cells to 5-aza-dC in our
CRISPR screen in WT HAP1 cells (Fig. 1B) and inhibition of PARP1/
2 has been reported to compromise BER of DNA lesions induced by
5-aza-dC (Orta et al, 2014). To test the role of DCTD in this context,
we inactivated the SSB repair factor PARP1 in both WT and *DCTD*
KO HAP1 cells using CRISPR/Cas9 and observed increased sensitivity
of PARP1-depleted cells to 5-aza-dC in both backgrounds relative to
cells transduced with the empty vector (Fig. EV1H–J). This suggested
that SSBs arising from BER of dU (Requena et al, 2016), 5-aza-dC
and/or 5-aza-dU incorporated into genomic DNA can contribute to
DCTD-independent 5-aza-dC cytotoxicity. However, given that no
DNA glycosylases scored as resistance hits in our 5-aza-dC CRISPR
screen (possibly suggesting the involvement of more than one
glycosylase), the mechanism by which these SSBs are generated
remains unclear.

Given our finding that DCTD drives a mechanism of 5-aza-dC
cytotoxicity that is independent of DNMT1-DPCs, we reasoned

that in *DCTD* KO cells, 5-aza-dC cytotoxicity would be caused by other routes, such as DNMT1-DPC formation. To gain insights into DCTD-independent mechanisms of 5-aza-dC sensitivity and resistance, we performed a genome-scale CRISPR screen in *DCTD* KO cells. Strikingly, in contrast to our 5-aza-dC screen in WT cells (Fig. 1B), in *DCTD* KO cells, gRNAs targeting *TYMS*, *DCTD* or *DCTPP1* no longer conferred 5-aza-dC resistance, while gRNAs against *DCK* still conferred strong resistance as expected (Fig. 2C). In addition, we identified several factors whose loss caused strong 5-aza-dC sensitivity (Fig. 2C). Most prominently, we identified PIAS4 and RNF4, which both have established roles in the replication-independent repair of DPCs (Sun et al, 2020; Liu et al, 2021), and the deubiquitylating enzyme USP7, one of whose functions is to regulate the DPC protease SPRTN (Zhao et al, 2021; Valles et al, 2020). We additionally identified factors with unclear roles in 5-aza-dC tolerance such as the dual SUMO1 and ubiquitin E3 ligase TOPORS and the ubiquitin E2 conjugating enzyme UBE2K (Fig. 2C). This indicates that in the absence of 5-aza-dC deamination, the relative contribution of DNMT1-DPCs to 5-aza-dC-induced cytotoxicity substantially increases.

## TOPORS is recruited to DNMT1-DPCs and promotes DPC tolerance

Considering the above findings, we speculated that in addition to identifying known DPC repair factors such as RNF4 and PIAS4, our CRISPR screen for 5-aza-dC sensitivity in *DCTD* KO cells may have uncovered as-yet unrecognised DPC repair factors. To explore this, we combined iPOND (Sirbu et al, 2011) with 5-aza-dC treatment to determine the proximal proteome of DNMT1-DPCs. Importantly, although 5-aza-dC is incorporated into nascent DNA by the replisome, our assay detects interactors of subsequently and postreplicatively formed DNMT1-DPCs. Briefly, HeLa TREx cells were synchronised via double-thymidine block and released into early/mid S-phase. Cells were co-treated with EdU and 5-aza-dC for 30 min to ensure their co-incorporation into nascent DNA, allowing subsequent specific isolation of DPC-containing chromatin (Fig. 3A). To this end, cells were crosslinked with formaldehyde followed by biotinylation of EdU through a click reaction with biotin-azide, DNA shearing and streptavidin-bead-mediated retrieval of DPC-containing DNA fragments. To validate our experimental protocol, we analysed the flowthrough and iPOND samples by western blotting with antibodies for DNMT1, histone H3 and tubulin (Fig. 3B). Histone H3, but not tubulin, was retrieved on streptavidin beads when cells were treated with EdU, indicating successful purification of nascent chromatin. Similarly, DNMT1 was detected on nascent chromatin, consistent with its key role in the maintenance of DNA methylation. Importantly, when cells were additionally treated with 5-aza-dC, the DNMT1 signal increased in iPOND samples, while histone H3 levels remained unchanged, indicating the formation of persistent DNMT1-DPC crosslinks in postreplicative chromatin (Fig. 3B).

Next, we set out to determine the proximal proteome of DNMT1-DPCs by using liquid chromatography with tandem mass tag (TMT)-multiplexed mass spectrometry. To recapitulate the successive stages of SUMO- and ubiquitin-dependent DNMT1-DPC repair (Borgermann et al, 2019; Sun et al, 2020; Liu et al, 2021; Weickert et al, 2023) (Fig. 3C), we compared proteins identified in iPOND samples of untreated cells with those of cells treated with 5-

aza-dC, co-treated with both 5-aza-dC and the SUMO E1 inhibitor (SUMO E1i) ML-792, or co-treated with 5-aza-dC and the ubiquitin E1 inhibitor (Ub E1i) TAK-243. Upon 5-aza-dC treatment, DNMT1, DNMT3A and DNMT3B, UHRF1, SUMO2 and SUMO1 were among the most enriched proteins in iPOND samples (Fig. 3D). We also identified factors previously shown to be involved in DPC repair such as the SUMO E3 ligase PIAS4 (Sun et al, 2020; Liu et al, 2021), VCP/p97 (Weickert et al, 2023; Noireterre et al, 2023; Fielden et al, 2020) and most proteasome subunits (Figs. 3D and EV2A; no peptides for RNF4 were detected in any of our samples). Inhibition of SUMOylation abrogated the recruitment of proteins involved in SUMO conjugation, including PIAS1-4, and TOPORS as well as VCP/p97 and to some extent the proteasome while also reducing basal levels of UBE2I (UBC9), but retained DNMT1 enrichment (Figs. 3E and EV2B). In contrast, while inhibition of ubiquitylation completely abrogated the recruitment of VCP/p97 and proteasome subunits, it did not diminish recruitment of PIAS1-4 or TOPORS (Figs. 3F and EV2C,D). Multiple other genome stability-associated proteins were enriched upon 5-aza-dC in iPOND samples, including components of the BRCA1-A complex and proteins involved in Topoisomerase II regulation (Fig. EV2D), possibly reflective of complex downstream responses to 5-aza-dC involving repair of peptide adducts left behind after DNMT1-DPC degradation.

To uncover proteins that are recruited to DNMT1-DPCs and whose loss causes 5-aza-dC sensitivity, we compared proteins identified to be in proximity of DNMT1-DPCs by iPOND with the top 2.5% of sensitivity hits in our 5-aza-dC CRISPR screen in *DCTD* KO cells after plotting the scores of all mutually occurring hits. This analysis identified the histone methyltransferase SETDB1 and its associated factor ATF7IP, the SUMO ligases PIAS2 and PIAS4, the VCP/p97 adaptor UBXN7 and the interstrand DNA crosslink (ICL) repair protein FANCA (Fig. 3G, lower right quadrant). In addition, this analysis highlighted the dual ubiquitin and SUMO1 E3 ligase TOPORS (Weger et al, 2005; Shinbo et al, 2005; Pungaliya et al, 2007; Rajendra et al, 2004). Furthermore, our iPOND data indicated that TOPORS is recruited to DNMT1-DPCs in a SUMO-dependent but ubiquitin-independent manner (Figs. 3C,E,F and EV2D). Together, these findings pointed to a direct and important role for TOPORS in response to 5-aza-dC-induced DNMT1-DPCs.

## TOPORS acts downstream of DNMT1-DPC SUMOylation to counter 5-aza-dC toxicity

To explore TOPORS functions, we established *TOPORS* KO clones of human *TP53* KO RPE1 cells. These cells displayed hypersensitivity towards 5-aza-dC when compared to isogenic controls (Figs. 4A and EV3A,B), a phenotype that we also observed in commercially available *TOPORS* KO HAP1 cells (Figs. 4B and EV3C). Furthermore, siRNA-mediated depletion of TOPORS also caused 5-aza-dC hypersensitivity in human U2OS cells (Figs. 4C and EV3D,E). In addition, we profiled *TOPORS* HAP1 cells for other hypersensitivity phenotypes relevant to genome stability, and observed hypersensitivity towards formaldehyde and camptothecin, which induce DPCs, but not to ionising radiation, which causes DNA breaks (Figs. 4D and EV3F–J). To connect the 5-aza-dC hypersensitivity phenotypes of TOPORS-deficient cells to DNMT1-DPCs, we depleted DNMT1 by using siRNA in WT and *TOPORS* KO HAP1 cells and found that DNMT1

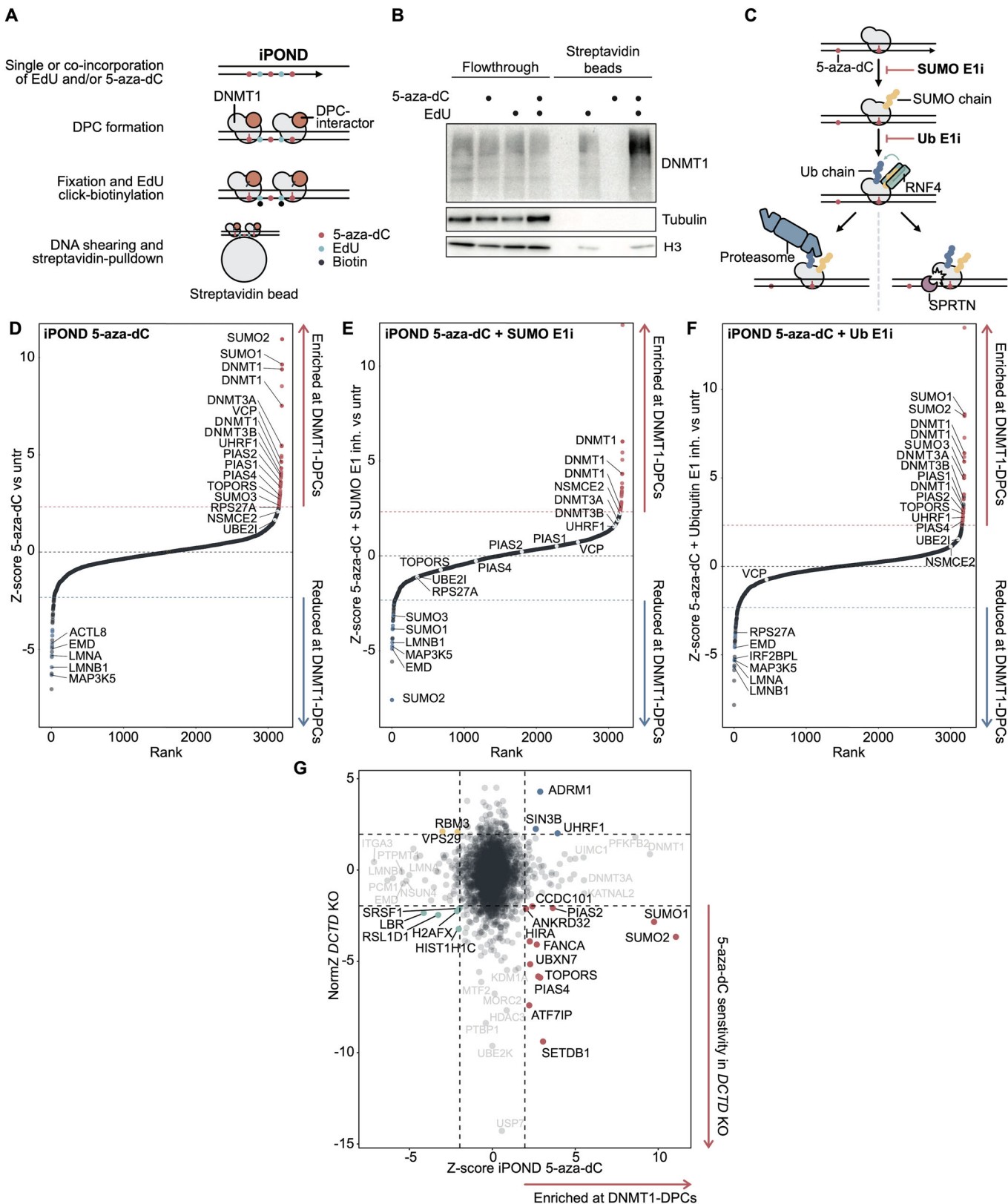

**Figure 3. iPOND identifies SUMO- and ubiquitin-dependent DNMT1-DPC-proximal factors.**

(A) Schematic outlining the iPOND approach. (B) HeLa TREx cells were treated with 5-aza-dC (10 μM), EdU (10 μM) or both and processed as depicted in (A). Samples were analysed by western blot using the indicated antibodies. Brightness and contrast was adjusted globally in ImageLab (Bio-Rad, version 5.2) to help visualise bands. Unprocessed blots are provided with the source data. Blots are representative of three independent experiments. (C) Schematic outlining global-genome (GG-) DPC repair and the impacts of SUMO or ubiquitin E1 inhibitors on this pathway. (D) Ranked standardised enrichment of proteins detected by iPOND-MS from 5-aza-dC treated over untreated cells. Dotted lines indicate thresholds of ±2.326. Proteins with an FDR ≤1% are represented by red or blue dots. Four replicates were measured. (E) Ranked standardised enrichment of proteins detected by iPOND-MS from co-treatment with 5-aza-dC and SUMO E1i over untreated cells. Dotted lines indicate thresholds of ±2.326. Proteins with an FDR ≤1% are represented by red or blue dots. Four replicates were measured. (F) Ranked standardised enrichment of proteins detected by iPOND-MS from co-treatment with 5-aza-dC and Ub E1i over untreated cells. Dotted lines indicate thresholds of ± 2.326. Proteins with an FDR ≤1% are represented by red or blue dots. Four replicates were measured. (G) Scatter plot comparing standardised enrichment scores of iPOND-MS from (D) with NormZ score of the top 2% of hits from our 5-aza-dC CRISPR/Cas9 screen in *DCTD* KO cells from Fig. 2C. Dotted lines indicate thresholds of ±2.326. Source data are available online for this figure.

depletion restored 5-aza-dC tolerance of *TOPORS* KO cells to WT levels (Figs. 4E,F and EV4A).

To build on these findings, and to validate our iPOND datasets, we established cell lines stably expressing HA-tagged TOPORS or containing empty vector (EV) as a control in U2OS cells also constitutively expressing green fluorescent protein (GFP)-tagged DNMT1 (U2OS GFP-DNMT1). We synchronised these cells in S-phase by single thymidine block and observed recruitment of TOPORS to DNMT1 upon release into 5-aza-dC treatment, as assessed by Proximity Ligation Assay (PLA) (Fig. 4G,H; see Fig. EV4B for representative images in U2OS GFP-DNMT1 HA-EV cells). In agreement with our previous observations, co-treatment of 5-aza-dC with SUMO E1i but not Ub E1i returned the PLA signal to background levels, indicating abolition of TOPORS recruitment to DNMT1-DPCs upon inhibition of SUMOylation but not of ubiquitylation (Fig. 4G,H). TOPORS is a known SUMO interactor (González-Prieto et al, 2021), suggesting that recruitment of TOPORS to DNMT1-DPCs could be mediated by direct interactions with SUMO chains formed on DPCs. Consistent with this idea, immunoprecipitation of WT GFP-TOPORS from 5-aza-dC-treated HeLa cells revealed robust interactions between GFP-TOPORS and high molecular weight SUMOylated proteins, when compared to the deoxycytidine treatment (Fig. 4I). Moreover, an interaction between GFP-TOPORS and heavily modified DNMT1 was strongly induced by 5-aza-dC treatment, likely corresponding to extensively SUMOylated 5-aza-dC-induced DNMT1-DPCs (Borgermann et al, 2019) (Fig. 4I). Notably, and in agreement with our iPOND and PLA data, co-treatment of GFP-TOPORS-expressing HeLa cells with SUMO E1i alongside 5-aza-dC essentially abrogated GFP-TOPORS' interactions with SUMOylated proteins and DNMT1 (Fig. 4I). Collectively, these data corroborate a direct role for TOPORS in response to 5-aza-dC-induced DNMT1-DPCs that entails its SUMO-dependent recruitment to DPCs.

## The RING domain and SUMO-interacting motifs of TOPORS mediate 5-aza-dC tolerance

To identify the functional modules of TOPORS that govern its role in 5-aza-dC tolerance, we next considered the domain structure of TOPORS. TOPORS contains a RING domain towards its N-terminus that mediates its ubiquitin ligase activity (Rajendra et al, 2004), a region that supports its SUMO1 ligase activity (Weger et al, 2005; Pungaliya et al, 2007), five canonical SUMO-interacting motifs (SIMs) (González-Prieto et al, 2021) and one putative, atypical SIM (aSIM; Fig. 5A). To further explore the functions of TOPORS, we established HAP1 *TOPORS* KO cell lines re-expressing HA-TOPORS$^{WT}$, a predicted ubiquitylation-defective

mutant (HA-TOPORS$^{CCAA}$) and a mutant bearing mutations in all six SIMs (HA-TOPORS$^{\Delta SIM}$) in a doxycycline-inducible manner (Fig. EV4C). In clonogenic survival assays, while HA-TOPORS$^{WT}$ expression in *TOPORS* KO cells restored 5-aza-dC tolerance substantially, this was not the case upon expression of HA-TOPORS$^{CCAA}$ or HA-TOPORS$^{\Delta SIM}$ (Figs. 5B,C and EV4D), suggesting that both TOPORS' ubiquitin ligase activity and its SUMO-interaction motifs are required for 5-aza-dC tolerance.

To address the role of TOPORS' ubiquitin ligase activity in 5-aza-dC tolerance, we immunoprecipitated GFP-TOPORS$^{CCAA}$ from HeLa cells, which, like GFP-TOPORS$^{WT}$, was proficient for 5-aza-dC-inducible interactions with modified DNMT1 and SUMOylated proteins (Fig. 5D). However, while we also detected an interaction between GFP-TOPORS$^{WT}$ and a high molecular weight ubiquitylated interactor, this was not overtly 5-aza-dC-inducible and was undetectable in co-immunoprecipitates of GFP-TOPORS$^{CCAA}$ (Fig. 5D). Reasoning that this high molecular weight ubiquitylated co-immunoprecipitate might represent autoubiquity-lated TOPORS, we performed more stringent immunoprecipita-tions to assess modifications on GFP-TOPORS itself. With this approach, we detected ubiquitylation of GFP-TOPORS$^{WT}$ that, consistent with our previous results, was not 5-aza-dC-inducible, while these high molecular weight ubiquitin signals were almost undetectable on GFP-TOPORS$^{CCAA}$ (Fig. 5E). Furthermore, high molecular weight SUMO conjugates were detectable on both GFP-TOPORS$^{WT}$ and GFP-TOPORS$^{CCAA}$, but this SUMOylation was not inducible by 5-aza-dC (Fig. 5E), possibly explaining the 'back-ground' SUMO signals in dC-treated GFP-TOPORS immunopre-cipitates in other experiments (such as in Fig. 5D). Together, these findings suggested that mutations rendering TOPORS ubiquitylation-defective compromise cellular tolerance of 5-aza-dC.

Considering that inhibition of SUMOylation using the SUMO E1i ML-792 abrogated TOPORS' recruitment to DNMT1-DPCs (Fig. 4G–I), we considered it likely that HA-TOPORS$^{\Delta SIM}$ failed to restore 5-aza-dC tolerance in *TOPORS* KO cells due to defective recruitment to DNMT1-DPCs. Indeed, immunoprecipitation experiments in HeLa cells showed that compared to GFP-TOPORS$^{WT}$, GFP-TOPORS$^{\Delta SIM}$ did not appreciably interact with high molecular weight conjugates or modified DNMT1 after 5-aza-dC treatment (Fig. 5F). Furthermore, in PLA experiments with U2OS cells co-expressing GFP-DNMT1 and either HA-TOPORS$^{WT}$ or HA-TOPORS$^{\Delta SIM}$, PLA signal induction between HA-TOPORS$^{\Delta SIM}$ and GFP-DNMT1 upon 5-aza-dC treatment was noticeably reduced compared with that observed between GFP-DNMT1 and HA-TOPORS$^{WT}$ (Fig EV4F,G), consistent with impaired recruitment of HA-TOPORS$^{\Delta SIM}$ to DNMT1-DPCs.

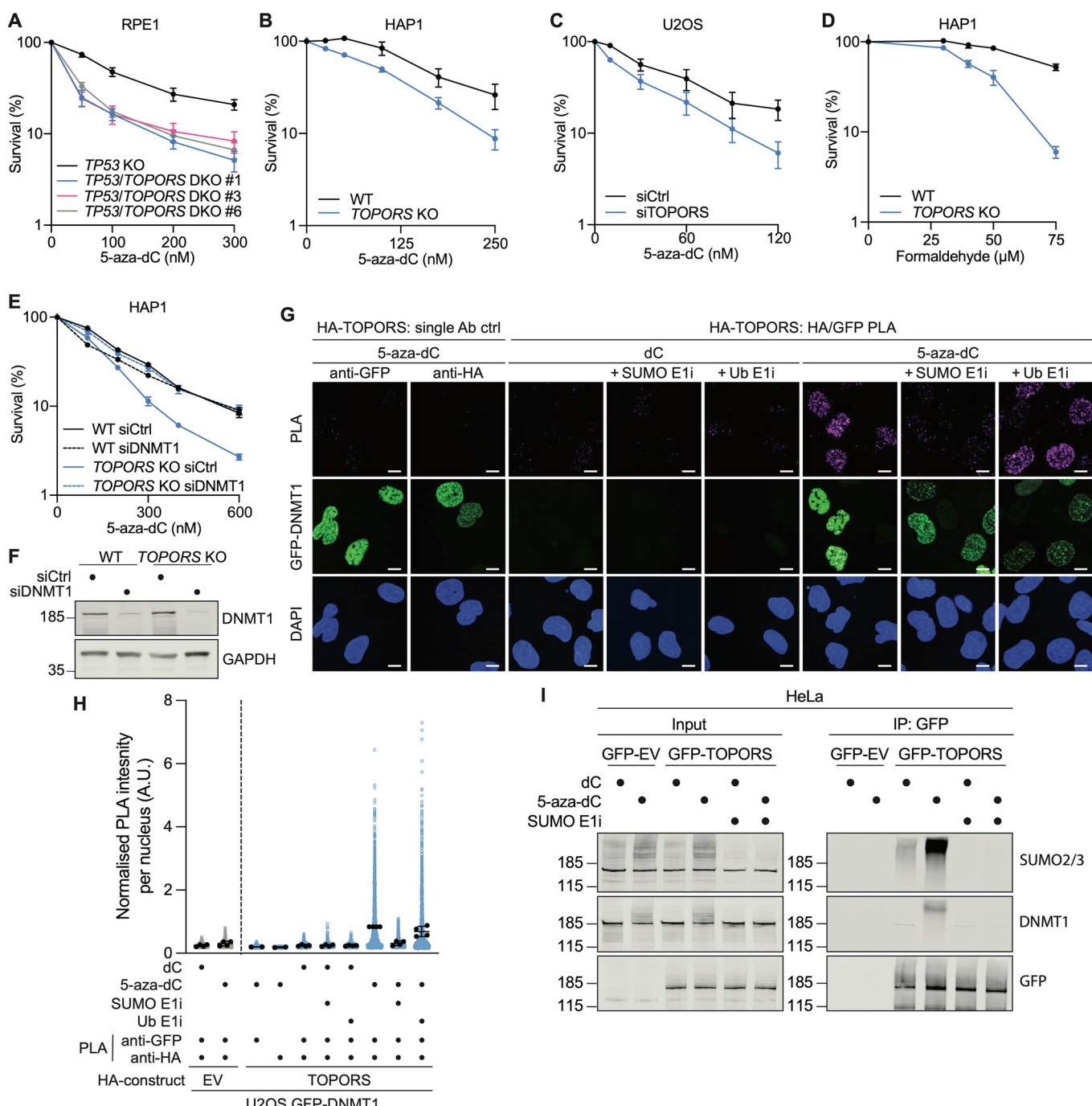

**2404** *The EMBO Journal* Volume 43 | Issue 12 | June 2024 | 2397 – 2423

Overall, these findings suggest that TOPORS functions in 5-aza-dC tolerance in a manner that is dependent on both its ubiquitin ligase activity and its recruitment to SUMOylated DNMT1-DPCs.

## TOPORS acts in parallel to RNF4 and UBE2K in mediating DNMT1-DPC tolerance

To better understand the role of TOPORS in promoting cellular tolerance to DNMT1-DPCs and thereby towards 5-aza-dC, we performed a further 5-aza-dC CRISPR screen in *TOPORS* KO HAP1 cells. This revealed that *TOPORS* KO cells are strongly sensitised to 5-aza-dC by additional loss of PIAS4, RNF4 or the ubiquitin-conjugating E2 enzyme UBE2K (Fig. 6A). Our CRISPR screen thus suggested that TOPORS acts independently of the PIAS4-RNF4 axis for DNMT1-DPC repair. Indeed, siRNA-mediated depletion of RNF4 caused dramatic sensitisation of *TOPORS* KO HAP1 cells to 5-aza-dC (Figs. 6B and EV5A,B), while siTOPORS caused profound additional sensitivity in *RNF4* KO HeLa cells (Figs. 6C and EV5C). Furthermore, cellular fitness, measured by proliferation rate, was severely compromised by the loss of both TOPORS and RNF4 in HeLa cells (Fig. 6D,E) and we were unable to knock out *TOPORS* in *RNF4* KO HeLa cells,

**Figure 4. TOPORS is a SUMO-dependent DPC tolerance factor.**

(A) Clonogenic survival assays in *TP53* KO and three clonally-derived *TP53/TOPORS* DKO RPE1 cell lines treated with 5-aza-dC; $n = 4$ biological replicates, error bars ± SEM. (B) Clonogenic survival assays in WT and *TOPORS* KO HAP1 cells treated with 5-aza-dC; $n = 4$ biological replicates, error bars ± SEM. (C) Clonogenic survival assays in WT U2OS cells transfected with siCtrl or siTOPORS and treated with 5-aza-dC; $n = 3$ biological replicates, error bars ± SEM. (D) Clonogenic survival assays in WT and *TOPORS* KO HAP1 cells treated with formaldehyde; $n = 3$ biological replicates, error bars ± SEM. (E) Clonogenic survival assays in WT and *TOPORS* KO HAP1 cells transfected with indicated siRNAs and treated with 5-aza-dC; $n = 3$ biological replicates, error bars ± SEM. (F) Western blot against the indicated antibodies from siRNA-transfected cells in (E); representative of three independent experiments. (G) Proximity ligation assay in U2OS cells expressing GFP-DNMT1 and HA-TOPORS, released from a single thymidine block and treated with deoxycytidine (dC), 5-aza-dC and/or Ub E1i or SUMO E1i for 1 h before pre-extraction of non-chromatin-bound proteins and fixation; scale bars = 10 μm. (H) Quantification of per-nucleus mean PLA intensities from (G), normalised to the median from the 5-aza-dC-treated U2OS GFP-DNMT1/HA-TOPORS condition. Black dots display the median normalised PLA intensity of each biological replicate for each condition; for single antibody controls, $n = 2$ biological replicates with a line at the mean. For all other experimental conditions, $n = 4$ independent biological replicates, error bars ± SEM. (I) Co-immunoprecipitation of GFP from extracts of HeLa cells expressing GFP (EV) or GFP-TOPORS$^{WT}$, released from thymidine block into S-phase and treated with dC, 5-aza-dC and/or SUMO E1i for 1 h, followed by western blotting for indicated proteins; representative of three independent experiments. Source data are available online for this figure.

possibly reflecting an inability of TOPORS- and RNF4-deficient cells to tolerate endogenously arising DPCs.

Interestingly, our CRISPR screen in *TOPORS* KO cells indicated that inactivation of the ubiquitin E2 conjugating enzyme UBE2K also caused further sensitisation to 5-aza-dC (Fig. 6A). Indeed, we found that UBE2K loss caused 5-aza-dC hypersensitivity that was dramatically exacerbated by additional loss of TOPORS (Fig. EV5D–G). To explore the relationship between UBE2K and RNF4, we depleted RNF4 using siRNA in *UBE2K* KO cells and in *UBE2K/TOPORS* double knockout (DKO) cells, which in both cases caused increased 5-aza-dC hypersensitivity (Fig. EV5H–K). These data indicated that UBE2K plays a role in 5-aza-dC tolerance that is independent of both TOPORS and RNF4. In support of this, despite being detected in the vicinity of nascent DNA in our iPOND experiments, UBE2K—in contrast to TOPORS—was not further enriched upon DNMT1-DPC induction by 5-aza-dC (Datasets EV2 and EV3). These findings place TOPORS in a DPC tolerance pathway acting alongside UBE2K and the RNF4 axis and suggest that TOPORS and RNF4 are two E3 ubiquitin ligases that perform at least partially overlapping functions necessary for survival.

While proteasomal degradation and SPRTN-dependent cleavage of DNMT1-DPCs are severely delayed in *RNF4* KO cells, they are not entirely abrogated (Weickert et al, 2023). Given that residual DNMT1-DPC proteolysis also depends on SUMOylation and ubiquitylation (Weickert et al, 2023), we hypothesised that TOPORS promotes polyubiquitylation and degradation of DNMT1-DPCs in parallel to RNF4. Thus, we employed the PxP assay (Weickert et al, 2023) (Fig. 2A) to examine the resolution of DNMT1-DPCs in WT and *RNF4* KO HeLa cells upon TOPORS depletion by siRNA. Accordingly, we synchronised cells in early/mid S-phase before DNMT1-DPCs' induction with a 30 min 5-aza-dC pulse (Fig. 6F). Following treatment, cells were either harvested directly or allowed to recover in drug-free medium for two or six hours. As shown previously (Weickert et al, 2023), DNMT1-DPCs were readily detectable following 5-aza-dC treatment but were swiftly lost in WT cells following washout of the drug (Fig. 6G; a proteolytic fragment arising from SPRTN-dependent cleavage of the DNMT1-DPC is highlighted with a red asterisk). In *RNF4* KO HeLa cells, DNMT1-DPC degradation was delayed and only observable at the six-hour time point. Strikingly, while siRNA-mediated TOPORS depletion in WT cells caused only a modest impairment in DNMT1-DPC degradation, TOPORS loss in *RNF4* KO cells virtually abolished DNMT1-DPC degradation at the time points tested (Fig. 6G). Corresponding deficiencies in DNMT1 degradation across these conditions were also visible in the PxP 'input' samples (Fig. 6G). Notably, the persistent DNMT1-DPCs

observable in *RNF4* KO cells further depleted of TOPORS were heavily modified with SUMO1 (Fig. 6G), an observation also made following stringent immunoprecipitation of GFP-DNMT1 from 5-aza-dC-treated U2OS GFP-DNMT1 cells, in which immunoprecipitated GFP-DNMT1 appeared to be more heavily modified with SUMO1 upon siRNA-mediated TOPORS depletion (Fig. 6H).

Together, our results support a model in which TOPORS is recruited to DNMT1-DPCs following their SUMOylation, where it promotes DPC polyubiquitylation and subsequent proteolysis in parallel to RNF4 to promote cell viability upon decitabine/5-aza-dC treatment.

# Discussion

In this study, we explored the cytotoxic effects of the hypomethylating agent 5-aza-dC by using unbiased genetic screens. We combined these insights with the first mapping of the proximal proteome of 5-aza-dC-induced DNMT1-DPCs. Our results shed light on two important modes of 5-aza-dC action. First, we reveal that a substantial amount of cytotoxicity originates from deamination of 5-aza-dC by DCTD and the generation of 5-aza-dUMP, highlighting DCTD loss as an important mechanism of decitabine resistance. Second, we find that the dual SUMO1 and ubiquitin E3 ligase, TOPORS, is a key player in global-genome DPC repair that promotes degradation of 5-aza-dC-induced DNMT1-DPCs, which is in agreement with recent reports that identify a central role for TOPORS in response to hypomethylating agents in MDS and AML models (Truong et al, 2022; Kaito et al, 2023).

Following the incorporation of 5-aza-dCTP into nascent DNA during S-phase and subsequent postreplicative DNMT1 trapping as a DPC, global-genome DPC repair acts to degrade DNMT1-DPCs through the action of both SPRTN and the proteasome (Borgermann et al, 2019; Liu et al, 2021; Weickert et al, 2023). This global-genome DPC repair is initiated upon DPC SUMOylation (Borgermann et al, 2019; Sun et al, 2020; Liu et al, 2021). Our data demonstrate that ensuing SUMO-dependent ubiquitylation is not only promoted by RNF4 but also by TOPORS. Indeed, loss of RNF4 compromises—but does not abolish—the SUMO-dependent degradation of DNMT1-DPCs (Liu et al, 2021; Weickert et al, 2023), which is only blocked upon loss of both TOPORS and RNF4, rendering cells extremely sensitive to 5-aza-dC. Like RNF4, TOPORS is recruited to SUMOylated DNMT1-DPCs, with the simultaneous depletion of both TOPORS and RNF4 being strongly detrimental to cell fitness. Therefore, TOPORS appears to act in

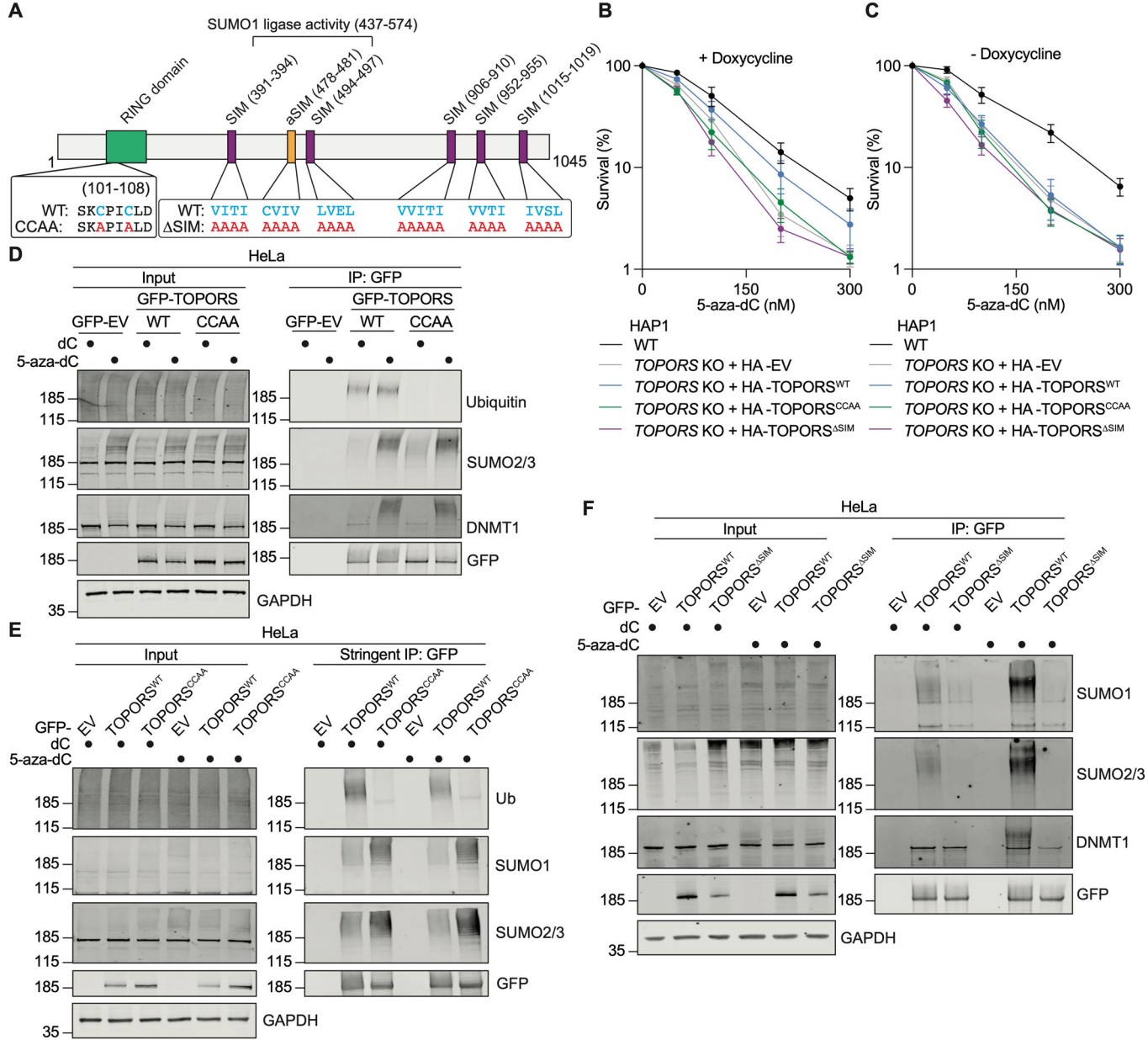

**Figure 5. TOPORS' RING domain and SUMO-interacting motifs mediate 5-aza-dC tolerance.**

(A) Domain map of TOPORS highlighting mutations made in the RING domain (CCAA) and of SIMs (ΔSIM). (B, C) Clonogenic survival assays in *TOPORS* KO cell lines with (B) or without (C) doxycycline-induced expression of the indicated forms of TOPORS, treated with 5-aza-dC; *n* = 4 biological replicates, error bars ± SEM. (D) Co-immunoprecipitation of GFP from extracts of HeLa cells expressing GFP (EV), GFP-TOPORS^WT or GFP-TOPORS^CCAA, released from thymidine block into S-phase and treated with dC or 5-aza-dC for 1 h, followed by western blotting for indicated proteins; representative of three independent experiments. (E) Co-immunoprecipitation and western blot as in (D) but under stringent, denaturing conditions; representative of three independent experiments. (F) Co-immunoprecipitation and western blotting as in (D) but with cells expressing GFP (EV), GFP-TOPORS^WT or GFP-TOPORS^ΔSIM; representative of three independent experiments. Source data are available online for this figure.

parallel to RNF4 to promote global-genome DPC repair with some level of mutual redundancy. However, this redundancy is likely limited. While RNF4 loss is embryonic lethal in mice (Hu et al, 2010), mutations in TOPORS are associated with a variant of retinitis pigmentosa (Chakarova et al, 2007) characterised by apoptotic rod cells, and with Joubert syndrome (Strong et al, 2023), a rare disease characterised by brain stem anomalies and in some

cases retinal dystrophy. In fruit flies, TOPORS was observed to localise to dedicated nuclear compartments where it regulates the activity of chromatin insulator complexes (Capelson and Corces, 2005). This implies that TOPORS and RNF4 have substantial non-overlapping functions in vivo. While subsequent studies will hopefully further explore and define the relationship between the two enzymes, it is conceivable that DNMT1-DPCs might be

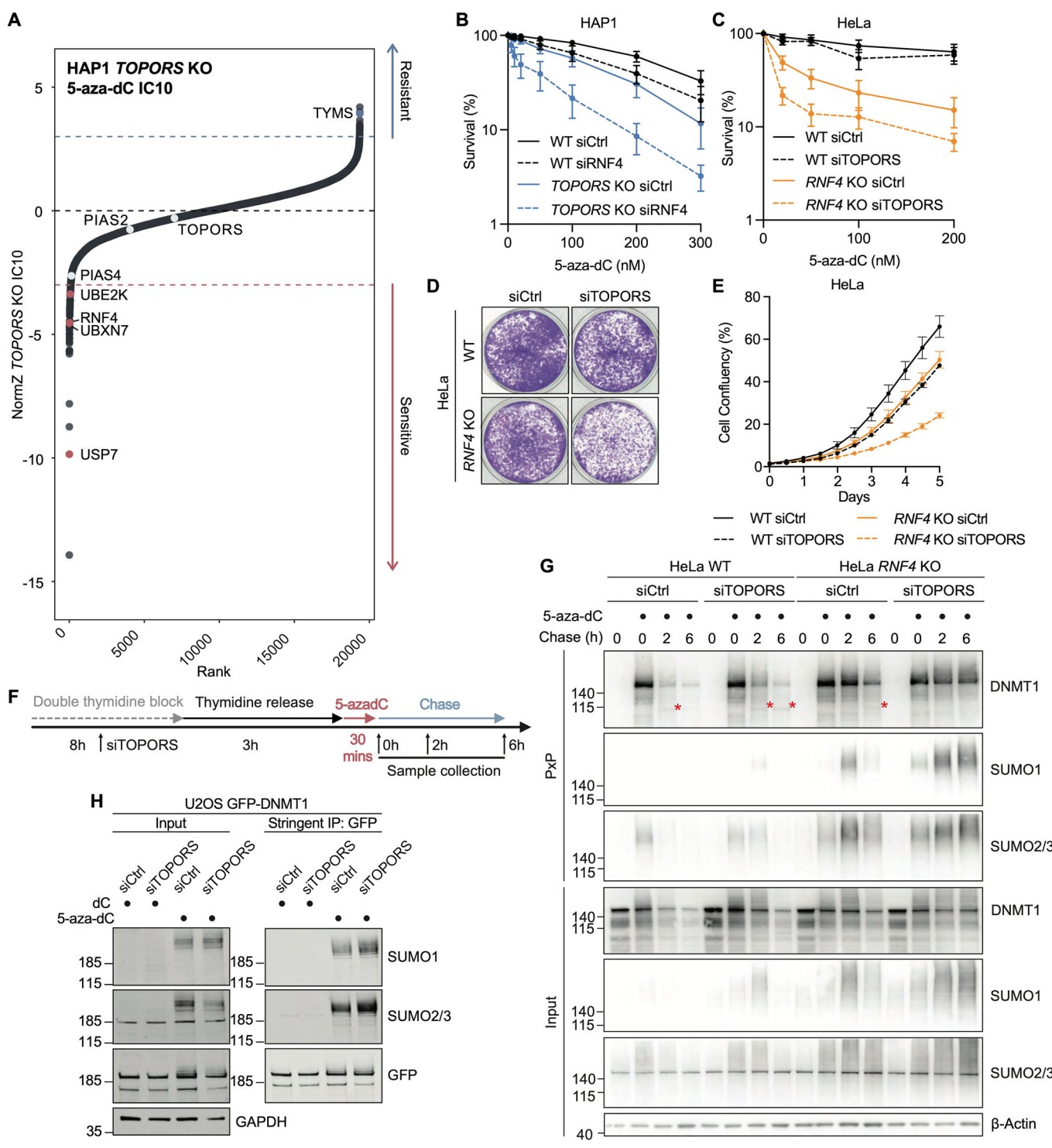

preferentially ubiquitylated by TOPORS or RNF4 depending on the chromatin context but can be modified by either if the lesion persists for a longer period of time. TOPORS is an unusual E3 ligase that promotes conjugation of both ubiquitin via its N-terminal RING domain, and SUMO1 via a region in its unstructured C-terminal tail. Our findings demonstrate that TOPORS is recruited to DNMT1-DPCs in a SUMO-dependent manner. However, whether SUMOylation is only required to

recruit TOPORS or also activates the enzyme, as in the case of RNF4, remains unclear (Plechanovová et al, 2011). In addition, understanding whether TOPORS' role in DPC repair requires its own SUMO E3 ligase activity or relies on other SUMOylating enzymes is an important future goal.

The formation of DNMT1-DPCs is particularly toxic upon loss of global-genome DPC repair factors, as exemplified by our observation that DNMT1 depletion rescues the 5-aza-dC

Figure 6. TOPORS and RNF4 operate in parallel to promote DPC degradation.

(A) Rank plot of a genome-wide CRISPR/Cas9 screen in *TOPORS* KO HAP1 cells treated with 5-aza-dC; dotted lines at NormZ scores of +3/−3 indicate thresholds for resistance and sensitivity hits, respectively. (B) Clonogenic survival assays in WT and *TOPORS* KO HAP1 cells transfected with siCtrl or siRNF4 and treated with 5-aza-dC; n = 4 biological replicates, error bars ± SEM. Note: two repeats in WT cells are shared with data shown in EV5H. (C) Clonogenic survival assays in WT and *RNF4* KO HeLa cells transfected with siCtrl or siTOPORS and treated with 5-aza-dC; n = 4 biological replicates, error bars ± SEM. (D) Representative image of cell confluency of WT and *RNF4* KO HeLa cells transfected with siCtrl or siTOPORS. (E) Cell confluency of WT and *RNF4* KO HeLa cells transfected with siCtrl or siTOPORS for a period of 5 days; data shown at mean ± SD, n = 3. Cell confluency was monitored using IncuCyte live cell imaging. (F) Treatment schematic for PxP assay in (G). (G) HeLa WT or *RNF4* KO were transfected with the indicated siRNAs, synchronised by double-thymidine block and treated with 5-aza-dC (10 μM) as depicted in (F). DNMT1-DPCs were isolated using PxP and analysed by western blotting using the indicated antibodies; representative of three independent experiments. Red asterisks (*) indicate SPRTN-dependent DNMT1 cleavage fragment. (H) Stringent immunoprecipitation of GFP-DNMT1 and subsequent western blotting from U2OS cells constitutively expressing GFP-DNMT1, transfected with siCtrl or siTOPORS and treated with dC or 5-aza-dC as indicated after release from a thymidine block into S-phase; representative of three independent experiments. Source data are available online for this figure.

hypersensitivity phenotype of *TOPORS* KO cells. However, our genetic screens identified an additional DNMT1-independent mode-of-action that is dominant in DPC repair-proficient HAP1 cells and originates from 5-aza-dCMP deamination to 5-aza-dUMP by DCTD. The generation of 5-aza-dUMP through deamination by DCTD has been shown previously (Chabot et al, 1983; Cashen et al, 2008; Almqvist et al, 2016), and 5-aza-dUMP has been suggested to perturb TYMS and lead to genomic uracil misincorporation (Almqvist et al, 2016; Requena et al, 2016), although to our knowledge we provide the first genetic evidence that DCTD underlies 5-aza-dC cytotoxicity through 5-aza-dUMP production. Notably, our CRISPR/Cas9 screens and survival assays with siRNA-mediated DNMT1 depletion indicate that even the residual 5-aza-dC sensitivity in *DCTD* KO cells is DNMT1-independent and appears to stem in part from 5-aza-dUMP generation. In *DCTD* KO but not WT cells, *TK1* loss causes 5-aza-dC resistance. This might suggest that in the absence of DCTD, the sequential action of CDA and TK1 on 5-aza-dC can generate 5-aza-dUMP and impart cytotoxicity. The different genetic dependencies of WT and *DCTD* KO cells upon 5-aza-dC treatment raises interesting questions about the mechanism of 5-aza-dUMP's cytotoxicity. We envision two possible scenarios, that are not necessarily mutually exclusive. First, given that 5-aza-dUMP interacts with TYMS in vitro and in cells (Almqvist et al, 2016; Cheung et al, 2023), it could inhibit the action of TYMS, as many other chemotherapeutics do (Tattersall et al, 1975; Rose et al, 2002). TYMS inhibition may be caused by the nitrogen substitution in place of a carbon at the 5' position within the pyrimidine ring of 5-aza-dUMP, which is expected to block its methylation by TYMS. The fact that in our CRISPR/Cas9 screens, loss of TYMS caused 5-aza-dC resistance in WT cells but had no effect in *DCTD* KO cells provides support for this scenario. Second, incorporation of 5-aza-dUTP, resulting from 5-aza-dUMP phosphorylation, into DNA could activate base excision repair (BER) and result in persistent DNA single-strand break (SSB) formation. This could be exacerbated by simultaneous inhibition of TYMS (Requena et al, 2016), decreasing dTTP levels and causing accumulation of endogenous dUTP, leading to increased uracil incorporation into DNA overall. Indeed, inhibition of the SSB repair factor PARP1 is known to synergise with 5-aza-dC (Orta et al, 2014; Muvarak et al, 2016) and we have found that PARP1 depletion sensitises both WT and *DCTD* KO cells to 5-aza-dC. However, while SSB induction by 5-aza-dC has been reported (Orta et al, 2014; Covey et al, 1986), it remains unclear whether these SSBs arise via BER action on genomic 5-aza-dCTP, 5-aza-dUTP, dUTP or all of these. Exploring

the relative contributions of these non-mutually exclusive scenarios is an exciting issue for future research.

Taken together, our findings may have important implications for the understanding and clinical applications of 5-aza-dC, and might serve as a starting point for the identification and development of new candidate biomarkers to guide patient stratification. Our results suggest that in the presence of proficient DPC repair, much of 5-aza-dC's cytotoxic effect originates not from DNMT1-DPCs but from the generation of 5-aza-dUMP. Given that hypomethylation is driven by DNMT1-DPC degradation, our work suggests that at clinically relevant doses, the origins of cytotoxicity are independent of DNA hypomethylation. This insight might help guide the development of new strategies to maximise either hypomethylation or cytotoxic effects depending on the clinical goal. Importantly, our data also highlight that the accepted mode of 5-aza-dC detoxification through deamination must be revisited.

## Cell culture

*Cell lines used in this study were maintained as detailed in Table EV1*
TOPORS was knocked out in *TP53* KO RPE1 cells stably expressing Cas9 (Serrano-Benitez et al, 2023) by transient transfection of sgRNA duplexes of an Alt-R tracrRNA (IDT) and a TOPORS-specific Alt-R crRNA (Table EV1), formed as per the manufacturer's recommendations and transfected using TransIT-LT1 (Mirus) as per the manufacturer's recommendations. 48 h later, cells were seeded at <1 cell/well in 96-well plates and expanded as clonal populations. To knock out *TOPORS* in *UBE2K* KO HAP1 cells, ribonucleoprotein (RNP) complexes were prepared between spCas9 (IDT) and the Alt-R sgRNA duplex prepared as described previously according to IDT's Alt-R protocol. RNPs were electroporated into HAP1 cells using a Neon NxT Electroporation System (ThermoFisher) as per the manufacturer's instructions. To validate candidate *TP53/TOPORS* DKO (double KO) and *UBE2K/TOPORS* DKO clones, genomic DNA extracts were prepared using the DNeasy Blood & Tissue Kit (Qiagen) as per the manufacturer's instructions. Clones were validated by TIDE analysis (Brinkman et al, 2014) following Sanger sequencing of PCR amplicons containing the targeted region of *TOPORS* using primers detailed in Table EV1.

To generate lentiviruses, HEK293T LentiX cells were transfected with TransIT-LT1 (Mirus) transfection reagent as well as a construct of interest (Table EV1) and the psPAX2 (Addgene 12260), and pMD2.G (Addgene #12259) constructs, according to the manufacturer's protocol. Forty-eight hours after transfection,

the viral supernatant was collected and filtered through a 0.45-µm sterile Millex-GP filter unit (Merck). The lentivirus was stored at −80 °C until use.

HAP1 WT and *DCTD* KO cell pools with PARP1 depletion by CRISPR/Cas9 were generated by infecting cells with lentiviral supernatant containing two different sgRNAs against PARP1 cloned into the BstB1 site in LentiCRISPRv2 (a gift from Feng Zhang (Sanjana et al, 2014); Addgene #52961), or infected with the empty vector. Transduced cells were selected using 1 µg/ml puromycin for 48 h, expanded and tested for PARP1 depletion by western blot.

U2OS cells stably expressing GFP-DNMT1 (U2OS GFP-DNMT1) were established by transfecting cells with pEGFP-DNMT1 using TransIT-LT1 (Mirus) as per the manufacturer's instructions, then selecting transfected cells with 1 mg/ml G418 (Gibco) for 7 days. G418-resistant cells were then seeded into 96-well plates at a concentration of 0.5 cells/well and after two weeks monoclonal cell lines were validated by GFP fluorescence and immunoblotting. U2OS GFP-DNMT1 cells were then infected with lentivirus-containing supernatants with pHA-EV-lentipuro or pHA-TOPORS-lentipuro and 48 h later selected with 2 µg/ml puromycin for 48 h. Puromycin-resistant polyclonal cell populations were then validated by qPCR and immunofluorescence against the HA-tag.

For the generation of *TOPORS* KO HAP1 cells expressing inducible constructs of HA-tagged TOPORS, 500,000 cells were seeded in media containing 8 µg/ml polybrene and transduced with lentiviral supernatant (prepared as described above) carrying the TET3G-T2A element. 48 h after transduction, cells were selected using 0.2 mg/ml hygromycin for 7 days. The selected population was then transduced with pTREG3-HA-EV, pTREG3-HA-TOPORS^WT, pTREG3-HA-TOPORS^CCAA, or pTREG3-HA-TOPORS^ΔSiM lentiviral supernatants. Cells were selected with puromycin (1 µg/ml) for 48 h and cultured in TET-negative FBS media to avoid background induction. When appropriate, expression of the inducible constructs was induced for 48 h at 1 µg/ml doxycycline, and validated by qPCR.

## CRISPR/Cas9 screens

In total, $2.5 \times 10^8$ WT HAP1, *DCTD* KO and *TOPORS* KO HAP1 cells were infected with the pre-packaged genome-wide All-in-One Brunello lentiviral library (Addgene 73179) at a multiplicity of infection (MOI) of 0.2. Following lentiviral integration, the transfected cells were selected with puromycin (0.6 µg/ml). Puromycin-resistant cells were passaged and expanded as required for 8 days. During this period, the cells were split into two independent replicates at a library coverage exceeding 500×, that is, a total of $50 \times 10^6$ cells per condition. After cell expansion, each replicate was treated with 5-aza-dC (30 nM for WT cells, 250 nM for *DCTD* KO cells and 12 nM for *TOPORS* KO cells) or DMSO, for a period of ten days. 5-aza-dC doses were predetermined based on pilot assays in untransduced cells seeded and passaged in screen-matched conditions with a range of 5-aza-dC concentrations. The first day of the ten-day treatment is Day 0 (T0) of the screen. During the course of the screen, cells were passaged and re-treated every other day, at a coverage of 500×.

For each replicate and treatment condition, cell pellets were collected on the first (T0) and last (T10) days of the screen, at a coverage of 500×. Upon harvesting of the cell pellets, genomic DNA was extracted using the QIAamp Blood Maxi kit (Qiagen).

Following the precipitation and purification of DNA in 70% ethanol, the DNA was dissolved to a concentration of 500 ng/µl in $H_2O$. The sgRNA sequences of each DNA sample were then amplified by PCR using the NEBNext® UltraTM II Q5® Master Mix (NEB) as well as a range of Illumina adaptor primers (Merck) flanking the sgRNA cassettes. Following the purification of the PCR products using the PCR column purification kit (Qiagen) as well as the QIAquick gel extraction kit (Qiagen), they were multiplexed using qPCR NEBNext library quant kit (E7630).

For sequencing and analysis, samples were multiplexed at 10 nM per lane and sequenced by next-generation sequencing on an Illumina NovaSeq 6000 instrument. A custom script (https://github.com/SimonLammmm/crispr_tools) was used to quantify sequencing reads and map them to the Brunello library. Median mapped counts per sample were $466 \pm 152$ (mean ± SD). On the mapped read counts, DrugZ software (Colic et al, 2019) was used to identify genes whose perturbation resulted in fitness effects indicated by a normalised Z score (NormZ) of a differential sample over a reference sample.

## Cell viability assays

For clonogenic survival assays, cells were seeded into 6-well plates at a concentration of 500–1500 cells/well in technical triplicate, with seeding density optimised to the cell line in question in order to account for baseline cell fitness defects where relevant (such as *TOPORS* KO cells with siRNF4). Twenty-four hours after seeding, cells were treated with the indicated drugs or ionising radiation (IR) at the specified doses; formaldehyde-treated cells were subjected to drug washout and supplied with fresh growth medium after 24 h of treatment. IR doses were performed using X-rays generated by a Rad Source RS 1800 Biological Irradiator. Six days after treatment, the surviving cells were fixed and stained with crystal violet. The number of colonies per well was counted, averaged between technical replicates and normalised to the number of colonies in untreated conditions. At least three biological replicates were performed per experiment, with each replicate displayed in quantification representing the mean normalised survival across three technical replicates.

For clonogenic survival assays of *TOPORS* KO HAP1 cells re-expressing inducible constructs of HA-tagged TOPORS, $5 \times 10^5$ cells were seeded per well in six-well plates and, 24 h later, exposed to 1 µg/ml doxycycline. The following day, clonogenic survival assays were seeded in media containing 1 µg/ml doxycycline and excess cells were re-plated into their original wells. Twenty-four hours after seeding, clonogenic assays were treated with 5-aza-dC and the re-plated excess cells were harvested for qPCR to confirm the expression of HA-tagged TOPORS.

For Incucyte proliferation assays, cells were transfected with the corresponding siRNAs as described below. The transfected cells were reseeded into six-well plates the following day (5000 of HeLa cells or 10,000 of U2OS T-Rex cells per well). Cell confluency was monitored and analysed using IncuCyte S3 live cell imaging system every 12 h for 5 days. After imaging, the cells were stained with crystal violet.

## Identification of proteins on nascent DNA (iPOND)

Overall, $5 \times 10^6$ HeLa TREx cells were seeded in two 15-cm dishes per condition and synchronised by a double-thymidine block. In

brief, cells were seeded in the morning and thymidine-containing media (2 mM, T9250) was added after 8 h. The next morning, cells were washed twice with 1× PBS and fresh, thymidine-free media was added for 9 h before re-adding thymidine-containing media and incubation overnight. Then, cells were washed twice with PBS, released into thymidine-free media and treated with EdU (10 µM) (Jena Bioscience, CLK-N001-100), 5-aza-dC (10 µM) (Sigma, A3656), Ub E1 inhibitor TAK-243 (1 µM) (Chemietek, AOB87172) or SUMO E1 inhibitor ML-792 (5 µM) (Axon Medchem, 3109) as indicated in figures. Treated cells were fixed in 10 ml 1% FA for 20 min at room temperature. Unreacted FA was quenched by the addition of 1 ml of 1.25 M glycine and incubation at room temperature for 10 min. Cells were scraped into 50-ml conical tubes, washed twice with PBS, snap-frozen in liquid nitrogen and stored at −80 °C until processed. Technical duplicates were pooled at this step.

iPOND was performed as described before (Sirbu et al, 2013). Briefly, pellets were resuspended in 1 ml permeabilization buffer (0.25% Triton X-100 in PBS) and incubated for 30 min at room temperature. The permeabilized cells were washed once with each 0.5% BSA in PBS and PBS prior to resuspending in 500 µl click reaction buffer (1×PBS, 10 µM biotin-azide, 100 mM sodium ascorbate, 100 mM copper sulphate). After 1 h incubation at room temperature, cells were washed once with each 0.5% BSA in PBS and PBS and resuspended in 1 ml RIPA lysis buffer (100 mM Tris pH 7.5, 150 mM NaCl, 1% IGEPAL, 0.1% SDS, 0.5% sodium deoxycholate, cOmplete EDTA-free protease inhibitor cocktail). The lysate was sonicated in an ultrasonicator (Covaris E220 *evolution*) for 10 min followed by centrifugation at $21,130 \times g$ for 10 min. The supernatant was incubated with 30 µl streptavidin-sepharose (Cytavia, 90100484) at 4 °C overnight. The next day, beads were washed with lysis buffer, 1 M NaCl and once more with lysis buffer before snap freezing in liquid nitrogen and storage at −80 °C.

## Identification of DNMT1-DPC proximal proteins by quantitative mass spectrometry

For quantitative mass spectrometry, beads were incubated for 1 h in 2 M Urea, 50 mM Tris-HCl (pH 7.5), 5 mM dithiothreitol containing trypsin. Beads were washed and the supernatant was saved. A reduction step with 5 mM dithiothreitol and an alkylation step with 20 mM chloroacetamide followed. Next, samples were digested overnight with trypsin. Samples were acidified using 1% formic acid and subsequently subjected to TMT labelling at 1.5:1 ratio for 1 h in 150 mM HEPES buffer (pH 8.5). TMT labelling was terminated with the addition of a 0.4% hydroxylamine solution, and excess labels were removed using reverse-phase Sep-Pak C18 cartridges. The TMT-labelled samples were pooled and desalted as previously described (Yu et al, 2021). Peptide fractions were analysed on a quadrupole Orbitrap mass spectrometer (Orbitrap Exploris 480, Thermo Scientific) equipped with a UHPLC system (EASY-nLC 1000, Thermo Scientific) as described (Michalski et al, 2011; Kelstrup et al, 2012). Peptide samples were loaded onto C18 reversed-phase columns (15 cm length, 75-µm inner diameter and 1.9-µm bead size) and eluted with a linear gradient from 8 to 40% acetonitrile containing 0.1% formic acid in 2 h. The mass spectrometer was operated in data-dependent mode, automatically switching between MS and MS2 acquisition. Survey full scan MS spectra (m/z 300–1650) were acquired in the Orbitrap. The 20 most intense ions were sequentially isolated and fragmented by higher-energy C-trap dissociation (HCD) (Olsen et al, 2007). Peptides with unassigned charge states, as well as with charge states less than +2 were excluded from fragmentation. Fragment spectra were acquired in the Orbitrap mass analyser. Raw data files were analysed using MaxQuant (development version 1.6.14.0) (Cox and Mann, 2008). Parent ion and MS2 spectra were searched against a database containing 98,566 human protein sequences obtained from UniProtKB (July 2021 release) using the Andromeda search engine (Cox et al, 2011). Spectra were searched with a mass tolerance of 6 ppm in MS mode, 20 ppm in HCD MS2 mode and strict trypsin specificity, allowing up to three missed cleavages. Cysteine carbamidomethylation was searched as a fixed modification, whereas protein N-terminal acetylation and methionine oxidation were searched as variable modifications. The dataset was filtered based on posterior error probability (PEP) to arrive at a false discovery rate (FDR) of less than 1% estimated using a target-decoy approach (Elias and Gygi, 2007).

For statistical analysis, MaxQuant output data were imported into R. Only proteins identified in all four replicates of each condition were kept for downstream analysis. Intensities were $\log_2$ transformed and quantile normalised between the replicates using the R package preprocessCore. Significantly enriched proteins were identified by employing a moderated t-test using the R package limma with Benjamini-Hochberg FDR correction.

## Proximity ligation assay (PLA)

U2OS GFP-DNMT1 HA-EV, HA-TOPORS[WT] or HA-TOPORS[ΔSIM] cells were seeded into 96-well imaging plates (PerkinElmer) at a density of $8 \times 10^4$ cells per well. The following day, growth media was changed to normal growth media containing 2 mM thymidine (Sigma-Aldrich) to arrest cells at the G1/S boundary. After 20–24 h, cells were washed 4× in warm PBS, then washed once in normal growth media and incubated for 5 min at 37 °C. Cells were then released into S-phase in normal growth media at 37 °C in the presence or absence of 1 µM Ub E1 inhibitor or 2 µM SUMO E1i (Medchem Express) for 30 min. Media was then exchanged for treatment media containing 10 µM 2'-deoxycytidine (dC; Sigma-Aldrich) or 5-aza-dC (Sigma-Aldrich) either with or without 1 µM Ub E1 inhibitor or 2 µM SUMO E1i and incubated at 37 °C for 1 h. Non-chromatin-bound proteins were cleared by pre-extraction in ice-cold 0.2% Triton X-100 on ice for 2 min, after which cells were washed once with PBS and fixed with 4% formaldehyde for 15 min. PLA was then performed using anti-HA and anti-GFP antibodies; details of the antibodies used in this study can be found in Table EV1. The Duolink In Situ PLA Anti-Mouse Minus and Anti-Rabbit Plus probes (Sigma-Aldrich) and Duolink In Situ Detection Reagents FarRed Kit (Sigma-Aldrich) were then used as per the manufacturer's instructions. Following PLA, cells were counter-stained with 1 µg/ml DAPI at room temperature for 2 min and then washed 4× with PBS. Images were acquired on a Zeiss 880 confocal microscope and analysis of PLA signal was performed using CellProfiler, or on a Revvity Opera Phenix and analysed using the integrated Harmony software. Per-nucleus mean PLA intensity was calculated and normalised to the median per-nucleus mean PLA intensity of the GFP-DNMT1/HA-TOPORS PLA in 5-aza-dC-treated cells.

## Immunoprecipitations

For immunoprecipitation of GFP-TOPORS, five 15-cm dishes of HeLa cells per condition were transfected with the 5 µg per dish of appropriate plasmids (plasmid details can be found in Table EV1) and TransIT-LT1 (Mirus) according to the manufacturer's instructions. The next day, cells were synchronised using 2 mM thymidine (Sigma-Aldrich). In all, 20–24 h later, cells were washed 4× with warm PBS and once in warm growth media and incubated for 5 min at 37 °C. Cells were then released into normal growth media for 30 min at 37 °C, then treated for 1 h with 10 µM dC or 5-aza-dC in the presence or absence of 2 µM SUMO E1i, as indicated. Cells were washed twice with PBS, harvested by scraping in PBS and centrifuged at $300 \times g$ for 3 min. Excess PBS was aspirated and cell pellets were snap-frozen on dry ice and stored at −80 °C ahead of further processing.

Cells were lysed on ice for 10 min in ice-cold IP150 buffer (50 mM Tris-HCl pH 8, 150 mM NaCl, 2 mM MgCl$_2$, 1% Triton X-100) supplemented with 500 U/ml benzonase (Sigma-Aldrich) and cOmplete EDTA-free protease inhibitor tablets (Roche). For stringent IPs, IP1000 buffer (50 mM Tris-HCl pH 8, 1 M NaCl, 2 mM MgCl$_2$, 1% Triton X-100) was used after cell lysis for lysate standardisation and subsequent wash steps. Cells were then sonicated using a chilled water bath sonicator for a total of 6 min with 30 s ON/OFF pulses and placed on a rotor wheel at 4 °C for 45 min. For stringent IPs, 5 M NaCl was added to each tube to a final concentration of 1 M NaCl, and samples were placed on a rotor wheel at 4 °C for 10 min. Lysates were then cleared by centrifugation in a bench-top microcentrifuge at full speed for 15 min at 4 °C and standardised using IP150 (or IP1000 for stringent IPs) buffer according to protein quantification by Bradford assay, and 5% of each lysate was taken aside as an input sample, prepared for western blot by boiling in 1X Laemmli buffer with 5% beta-mercaptoethanol and stored at −20 °C. GFP-trap agarose beads (Chromotek) were equilibrated by washing twice in IP150 (or IP1000 for stringent IPs) buffer with centrifugation at $0.3 \times g$ for 2 min at 4 °C. Equal amounts of equilibrated beads were added to each tube and samples were placed on a rotor wheel at 4 °C overnight. Samples were washed four times by centrifuging for 1 min at $0.3 \times g$ at 4 °C, aspirating the supernatant and resuspending with 1 ml ice-cold IP150 buffer. Immunoprecipitates were eluted by boiling at 95 °C for 5 min in 2× Laemmli buffer supplemented with 5% beta-mercaptoethanol.

## siRNA transfections

*Details of siRNAs used in this study can be found in Table EV1*
For PxP experiments, $3 \times 10^6$ cells were seeded in 10-cm dishes in the morning, and thymidine-containing media (2 mM, T9250, Sigma) was added after 8 h. After thymidine addition, 10 µl siRNA (20 µM) and 25 µl Lipofectamine RNAiMAX Transfection Reagent (13778075, Thermo Scientific) were each diluted in 500 µl Opti-MEM Medium. Following a 5 min incubation, siRNA and Lipofectamine RNAiMAX Transfection Reagent dilutions were mixed. After an additional 15 min, the transfection mix was added to cells. The next morning, thymidine media was removed and cells were washed twice with PBS, trypsinized, counted and split in 6-cm dishes. Thymidine media was added again in the evening after cell attachment. The next morning cells were washed twice with PBS

and released into normal media for 2 h before adding fresh media containing 5-aza-dC (10 µM). After 30 min incubation, 5-aza-dC containing media was removed, cells were washed with PBS and allowed to recover. Cells were scraped on ice at the indicated timepoints and the pellets were frozen at −80 °C.

For clonogenic survival assays, $5 \times 10^5$ cells were seeded per well in six-well plates, with transfection mixes of 2.5 µl siRNA (20 µM) and 10 µl Lipofectamine RNAiMAX each prepared in 125 µl Opti-MEM medium prior to mixing, incubation and addition to cells as described above. Clonogenic survival assays were seeded from transfected cells as described previously. Forty-eight hours after transfection, excess cells were re-plated into their original wells. Twenty-four hours later, clonogenic assays were treated with 5-aza-dC and the re-plated excess cells were harvested for western blot at 72 h post-transfection to confirm knockdown.

For immunoprecipitation, $4 \times 10^6$ cells per siRNA were plated in 15-cm dishes and reverse-transfected as described using transfection mixes of 20 µl siRNA (20 µM) and 80 µl Lipofectamine RNAiMAX, each in 1.5 ml Opti-MEM medium prior to mixing and addition to dishes as described above. Twenty-four hours after transfection, each 15-cm dish was split 1:2 to two 15-cm dishes to yield sufficient dishes for dC and 5-aza-dC treatment after thymidine block and release as described above.

## Purification of x-linked proteins (PxP)

PxP to isolate DNMT1-DPCs was performed as described before (Weickert et al, 2023). In brief, cells were harvested by scraping into ice-cold PBS and either snap-frozen or directly used for PxP. The cell pellet was resuspended in PBS at a concentration of $2.5 \times 10^4$ cells/µl. In total, 10 µl of the cell suspension was directly lysed in 1× LDS (NP0007, Thermo Scientific) as input samples. The remaining suspension was mixed with an equal volume of low-melt agarose (2% in PBS, 1613111, Bio-Rad) and directly cast into plug moulds. After solidification at 4 °C for 5 min the plugs were transferred into 1 ml of cold lysis buffer (1×PBS, 0.5 mM EDTA, 2% sarkosyl, cOmplete EDTA-free protease inhibitor cocktail (4693132001, Merck), 0.04 mg/ml Pefabloc SC (11585916001, Merck) and incubated on a rotating wheel for 4 h at 4 °C. For electroelution, plugs were placed into the wells of ten-well SDS-PAGE gels (12%, 1.5 mm Novex WedgeWell). Electroelution was carried out at 20 mA per gel for 60 min. After electroelution, plugs were transferred to 1.5-ml tubes and boiled with 40 µl 1×LDS sample buffer and 10 µl reducing agent at 95 °C for 20 min.

## Western blotting

Cell pellets were lysed in 10 mM Tris pH 7.5 and 2% SDS, quantified through Bradford assays (LifeTechnologies), according to the manufacturer's protocol, and stored at −20 °C. After boiling the standardised protein lysates in 1X Laemmli buffer with 5% beta-mercaptoethanol, 20 µg of the lysates were loaded on NuPage 4–12% gradient Bis-Tris gels (Invitrogen) and run at 150 V for 90 min. The resolved proteins were then transferred onto a nitrocellulose membrane, which was blocked for 1 h in 5% milk in TBS-T (0.1% Tween-20 in TBS) and analysed by standard immunoblotting using the antibodies listed in Table EV1. All western blot images shown were obtained using the LI-COR platform (Biosciences) or by chemiluminescence using the ChemiDoc MP imager (Bio-Rad).

## RNA extraction and RT-PCR

Total RNA was extracted from harvested cells using the RNeasy Mini Kit with on-column DNase digestion (Qiagen), according to the manufacturer's instructions. During the extraction protocol, RNase-free plastic ware and solutions were used. After RNA extraction, isolated RNA was quantified using a Nanodrop spectrograph, and stored at −80 °C. cDNA was reverse transcribed from 1 µg of RNA using SuperScript IV VILO Master Mix (Invitrogen), according to the manufacturer's instructions. cDNA was stored at −20 °C. qPCR was performed using 1 µl of cDNA, 10 µL of 2× Fast SYBR Green Master mix (Thermo Fisher Scientific) and 500 nM forward and reverse primers, in a final volume of 20 µL. Primers were designed and ordered to span an exon-exon junction of the target genes (Sigma-Aldrich) and are detailed in Table EV1. qPCR analysis was performed on an QuantStudio 5 Real-Time PCR Systems (Thermo Fisher Scientific), in technical triplicates. Gene expression changes were calculated using the 2−ΔΔCt method (Livak and Schmittgen, 2001).

## Cell cycle analysis by flow cytometry

In total, $1.5 \times 10^6$ cells were seeded in 6 cm dishes in the morning and allowed to adhere overnight. The next day, cells were treated with 1 µM 5-aza-dC for 3 h or 6 h or left untreated. After 2.5 h or 5.5 h cells were pulsed with 10 µM EdU for 30 min while reserving one dish of non 5-aza-dC-treated cells as an EdU negative control. Cells were then washed twice with PBS, trypsinised and collected in a 1.5-ml tube. Samples were spun down at $500 \times g$ in a tabletop centrifuge for 5 min, the supernatant discarded and the pellet was washed with 1 ml PBS. The resulting cell pellet was resuspended in 200 µl PBS containing 1x eFluor780 viability dye and incubated for 30 min at 4 °C in the dark. Subsequently, cells were washed with 1% BSA in PBS and fixed in 200 µl 4% formaldehyde in PBS for 15 min at room temperature.

For EdU click-labelling, fixed cells were resuspended in 250 µl 0.25% Triton-X100 in PBS and incubated for 20 min at room temperature. The permeabilized cells were washed once with 1 ml 1% BSA in PBS and the cell pellet was resuspended in 250 µl click-chemistry mix (11 mM sodium ascorbate, 39.5 mM Tris-HCl pH 8.0, 60 µM Alexa Fluor 488-azide, 4 mM CuSO$_4$, 10 nM DAPI in 1xPBS) and incubated in the dark at room temperature for 30 min while occasionally resuspending sedimented cells. Finally, cells were washed with 1 ml 1% BSA in PBS and resuspended in 200 µl 1% BSA in PBS.

For flow cytometry, samples were analysed on a BD LSRFortessa (BD Bioscience) equipped with 355/405/488/561/640 nm lasers with a minimum count of 10,000 events. Results were analysed using FloJo™ v10.7 software (BD Life Sciences). Staining with fixable viability dye eFluor780 was used to exclude dead cells and the mean fluorescence intensity in FITC channel was measured for single and live to classify cells into EdU negative and positive cells based on the FITC intensity of no EdU control samples.

## Experimental study design statement

For all experiments with the exception of CRISPR/Cas9 screens (which were performed in biological duplicate) and reagent validation experiments (which were conducted at least twice), a minimum of three independent observations yielding similar results were considered sufficient with respect to sample size. No blinding was done.

## Availability of biological materials

All newly established biological materials, such as plasmids and cell lines described in this study, are available from the corresponding authors upon reasonable request, subject to the establishment of a suitable Material Transfer Agreement (MTA) where relevant.

# Data availability

Mass spectrometry data have been deposited to the ProteomeXchange Consortium via the PRIDE partner repository (https://www.ebi.ac.uk/pride/) with the dataset identifier PXD045071. NormZ scores arising from DrugZ analysis of the three CRISPR/Cas9 screens presented in this study are provided in Dataset EV1; remaining DrugZ outputs and raw NGS counts are available from the corresponding authors upon reasonable request.

The source data of this paper are collected in the following database record: biostudies:S-SCDT-10_1038-S44318-024-00108-2.

# Peer review information

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

## Acknowledgements

The authors thank Dr. Marcin J Suskiewicz (CNRS Orléans, France), for helpful discussions and advice. Research in the lab of JS is funded by European Research Council (ERC Starting Grant 801750 DNAProteinCrosslinks), Alfried Krupp Prize for Young University Teachers awarded by Alfried Krupp von Bohlen und Halbach-Stiftung, European Molecular Biology Organization (YIP4644), The Vallee Foundation, and Deutsche Forschungsgemeinschaft (DFG, German Research Foundation)—Project-ID 213249687—SFB 1064 and Project-ID 393547839—SFB 1361. Research in the Beli lab is funded by the Deutsche Forschungsgemeinschaft (German Research Foundation, DFG): Project-ID BE 5342/3-1, Project-ID 393547839—SFB 1361, Project-ID 464588647—SFB 1551. The SPJ laboratory is supported by Cancer Research UK (CRUK) Discovery Award DRCPGM\100005, CRUK core grant A:29580 and ERC Synergy Award 855741 (DDREAMM). The lab was also supported by CRUK Programme grant C6/A18796 and core funding grants C6946/A24843 and WT203144 to the Gurdon Institute. CJC, MS-C, NG, AW, GZV and SR were funded by CRUK Programme grant C6/A18796 and CRUK Discovery Award DRCPGM\100005; CPC is funded by a CRUK studentship DRCPGM\100005; VG, AS-B, SL and GD'A by ERC Synergy Award 855741. SWA is funded by the Mark Foundation For Cancer Research ASPIRE II award ASP-II-4555793532. SPJ receives a salary from the University of Cambridge. We thank Nicola Lawrence and Kay Harnish of the Gurdon Institute Imaging and Scientific Facilities, at the Gurdon Institute, University of Cambridge, and the RICS and Light Microscopy core facilities of the CRUK Cambridge Institute, and Dr. Kate Dry for editorial assistance. For the

## Author contributions

**Christopher J Carnie**: Conceptualisation; Data curation; Supervision; Validation; Visualisation; Writing—original draft; Project administration; Writing—review and editing. **Maximilian J Götz**: Conceptualisation; Data curation; Formal analysis; Validation; Investigation; Visualisation; Writing—original draft; Writing—review and editing. **Chloe S Palma-Chaundler**: Data curation; Formal analysis; Validation; Investigation; Visualisation; Writing—original draft; Writing—review and editing. **Pedro Weickert**: Validation; Investigation; Visualisation; Writing—review and editing. **Amy Wanders**: Formal analysis; Investigation. **Almudena Serrano-Benitez**: Supervision; Validation; Investigation; Writing—review and editing. **Hao-Yi Li**: Investigation; Visualisation. **Vipul Gupta**: Formal analysis; Visualisation. **Samah W Awwad**: Investigation. **Christian J Blum**: Investigation; Methodology; Writing—review and editing. **Matylda Sczaniecka-Clift**: Investigation; Methodology. **Jacqueline Cordes**: Investigation. **Guido Zagnoli-Vieira**: Investigation; Methodology. **Giuseppina D'Alessandro**: Investigation; Methodology; Writing—review and editing. **Sean L Richards**: Investigation. **Nadia Gueorguieva**: Investigation. **Simon Lam**: Formal analysis; Writing—review and editing. **Petra Beli**: Supervision; Funding acquisition. **Julian Stingele**: Conceptualisation; Supervision; Funding acquisition; Writing—original draft; Project administration; Writing—review and editing. **Stephen P Jackson**: Supervision; Funding acquisition; Writing—original draft; Project administration; Writing—review and editing.

Source data underlying figure panels in this paper may have individual authorship assigned. Where available, figure panel/source data authorship is listed in the following database record: biostudies:S-SCDT-10_1038-S44318-024-00108-2.

## Funding

## Disclosure and competing interests statement

The authors declare no competing interests.

# Expanded View Figures

**Figure EV1.  DCTD promotes DNMT1-independent 5-aza-dC cytotoxicity.**

(A, B) Western blot in WT and *DCTD* KO (A) or *DCK* KO (B) HAP1 cells with the indicated antibodies; representative of 3 (A) and 2 (B) independent experiments. The DCTD-specific band in (A) is marked with a red asterisk (\*). (C) Speculative model for DCK-independent incorporation of 5-aza-dC into DNA and subsequent DNMT1 trapping. Briefly, upon cellular uptake, 5-aza-dC can be deaminated by CDA, followed by triphosphorylation involving the activity of TK1, followed by possible conversion of 5-aza-dUTP to 5-aza-dCTP by CTPS1 and subsequent DNA incorporation and DNMT1 trapping. (D) Western blot for the indicated antibodies in WT and *DCTD* KO HAP1 cells transfected with the indicated siRNAs; representative of three independent experiments. (E) Clonogenic survival assays with siRNA-transfected WT and *DCTD* KO cells from (D) treated with 5-aza-dC; $n = 3$ biological replicates, error bars ± SEM. (F) Representative images from (D) at selected 5-aza-dC doses. (G) Percentage of EdU-positive cells determined by flow cytometry of HAP1 WT, *DCTD* KO and *DCK* KO cells either untreated or treated with 1 μM 5-aza-dC for 3 h or 6 h; $n = 3$ biological replicates, error bars show mean ± SD. (H) Western blot with the indicated antibodies of polyclonal cell populations of WT and *DCTD* KO cells following CRISPR/Cas9-mediated depletion of PARP1 with the indicated sgRNAs; representative of two independent experiments. (I) Clonogenic survival assays on cells from (G) treated with 5-aza-dC; $n = 3$ biological replicates, error bars ± SEM. (J) Representative images from (I) at selected 5-aza-dC doses. Source data are available online for this figure.

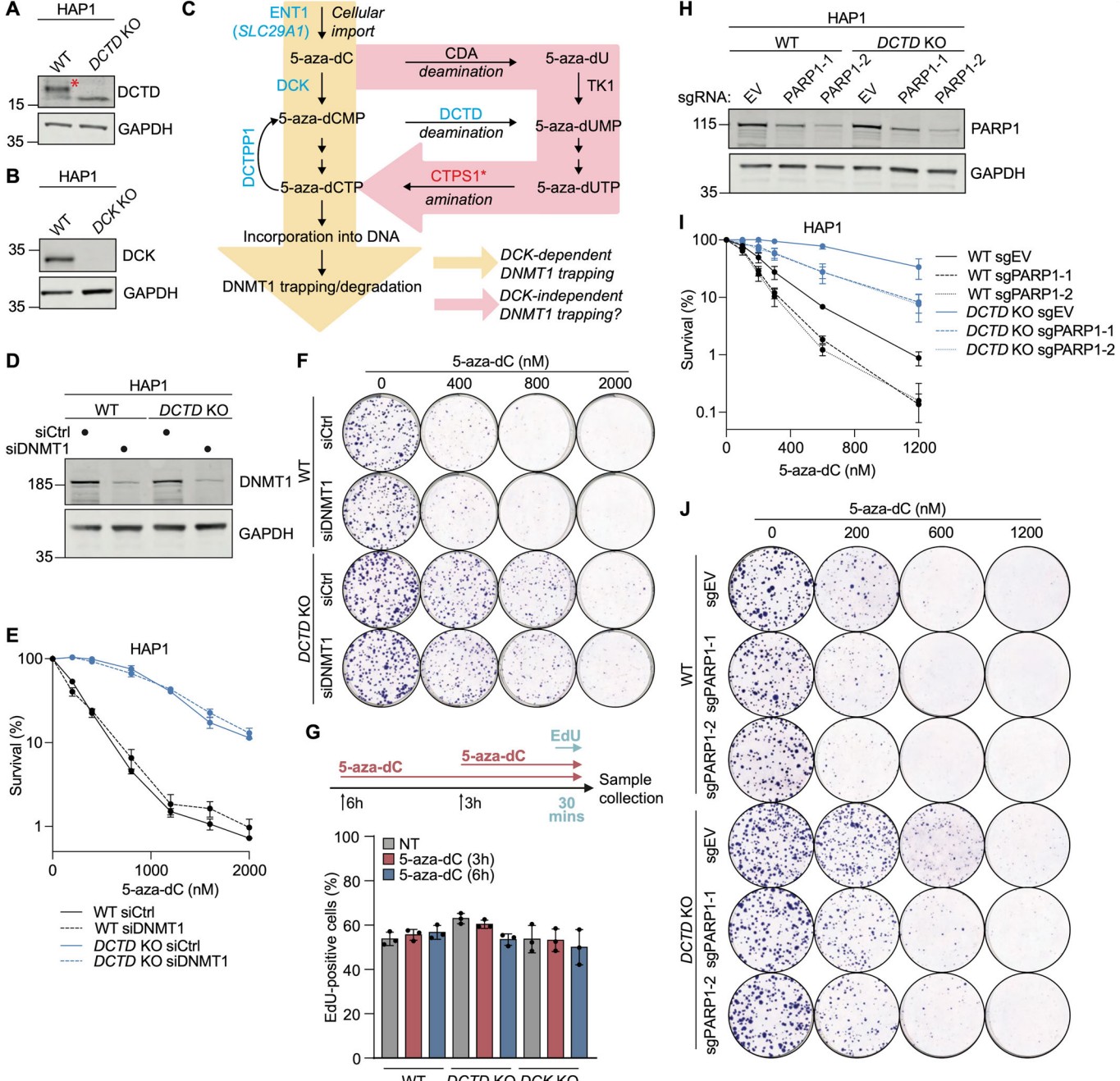

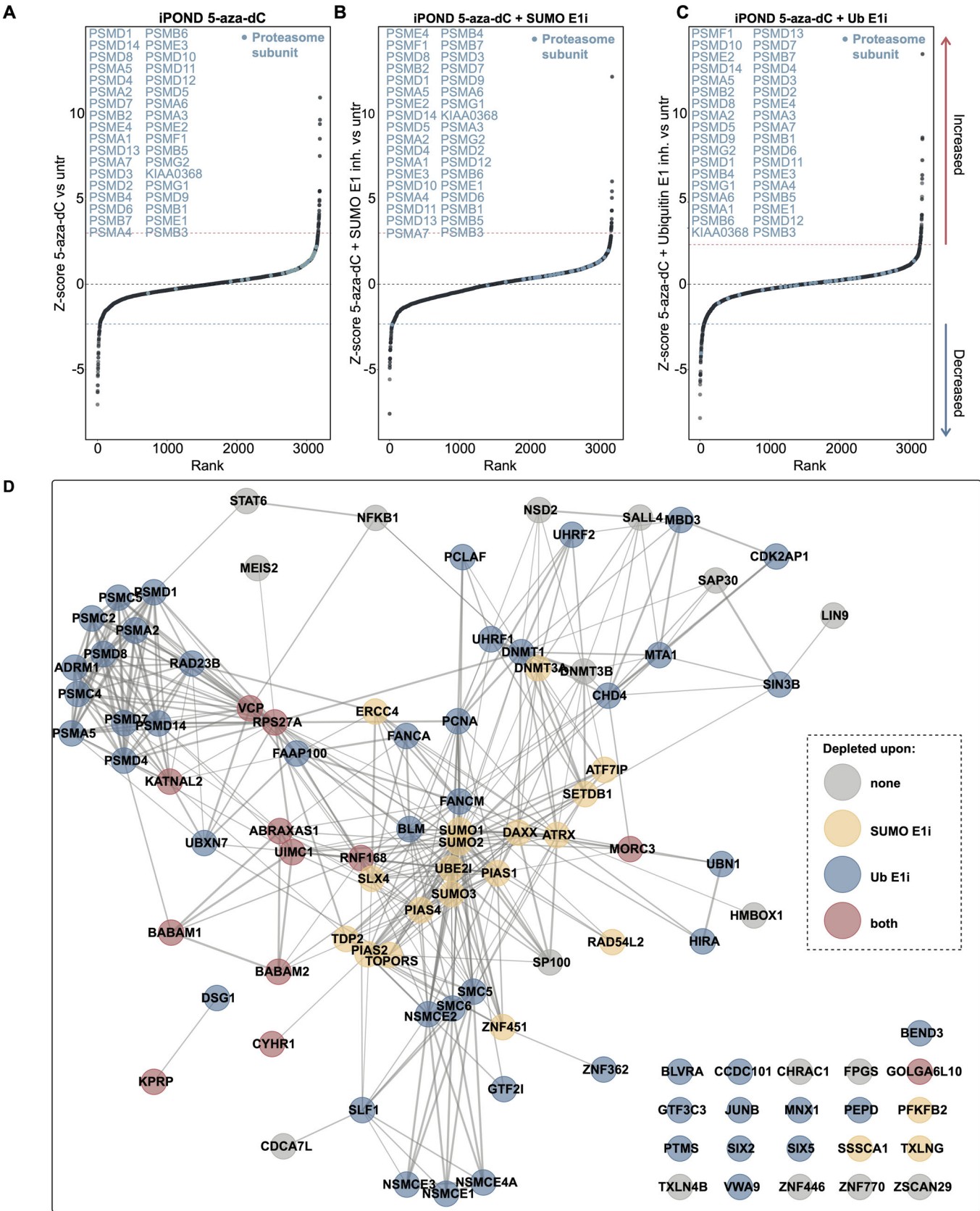

◀ **Figure EV2.   iPOND identifies the SUMO- and ubiquitin-dependent DNMT1-DPC-proximal proteome.**

(**A–C**) Ranked standardised enrichment of proteasomal subunits detected by iPOND-MS from 5-aza-dC-treated (**A**), 5-aza-dC- and SUMO E1i-co-treated (**B**), and 5-aza-dC- and Ub E1i-co-treated (**C**) over untreated cells. (**D**) STRING analysis of proteins enriched on nascent DNA after 5-aza-dC treatment with SUMO/ubiquitin dependencies indicated, as assessed by iPOND-MS.

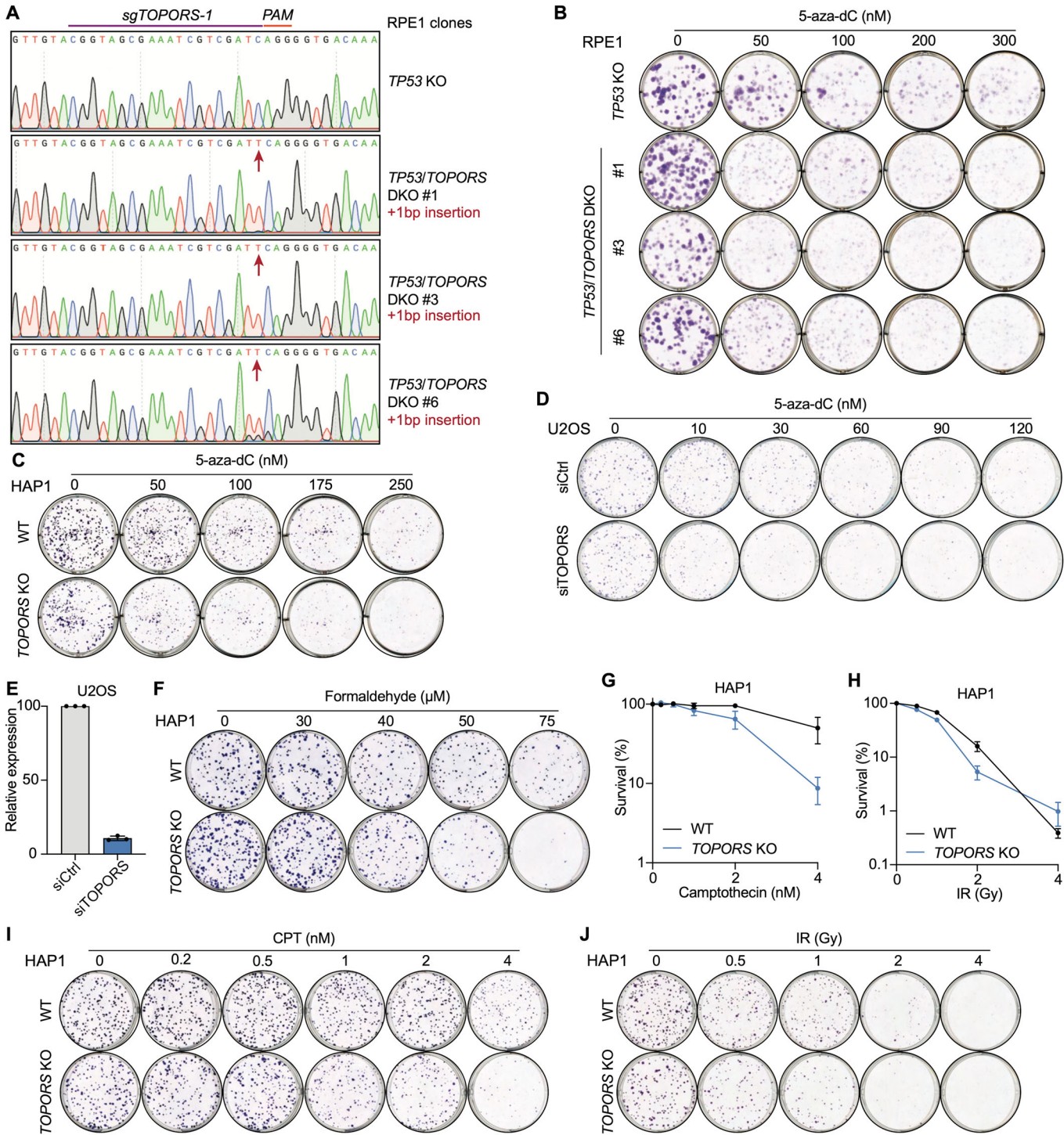

**Figure EV3. TOPORS loss sensitises cells to DPC-inducing agents.**

(A) Validation by Sanger sequencing of *TP53/TOPORS* DKO RPE1 clones. (B–D) Representative images of selected 5-aza-dC doses from clonogenic survival assays in Fig. 4A (B), Fig. 4B (C) and Fig. 4C (D). (E) Relative expression levels of TOPORS from U2OS cells 72 h after siRNA-mediated depletion of TOPORS measured by qPCR, relative to GAPDH expression and normalised to TOPORS expression level in siCtrl cells; $n = 3$ replicates, error bars ± SEM. (F) Representative images from clonogenic survival assays in Fig. 4D. (G, H) Clonogenic survival assays in WT and *TOPORS* KO HAP1 cells treated with camptothecin (G) and ionising radiation (IR; H); $n = 3$ biological replicates, error bars ± SEM. (I, J) Representative images from clonogenic survival assays in (G, H), respectively. Source data are available online for this figure.

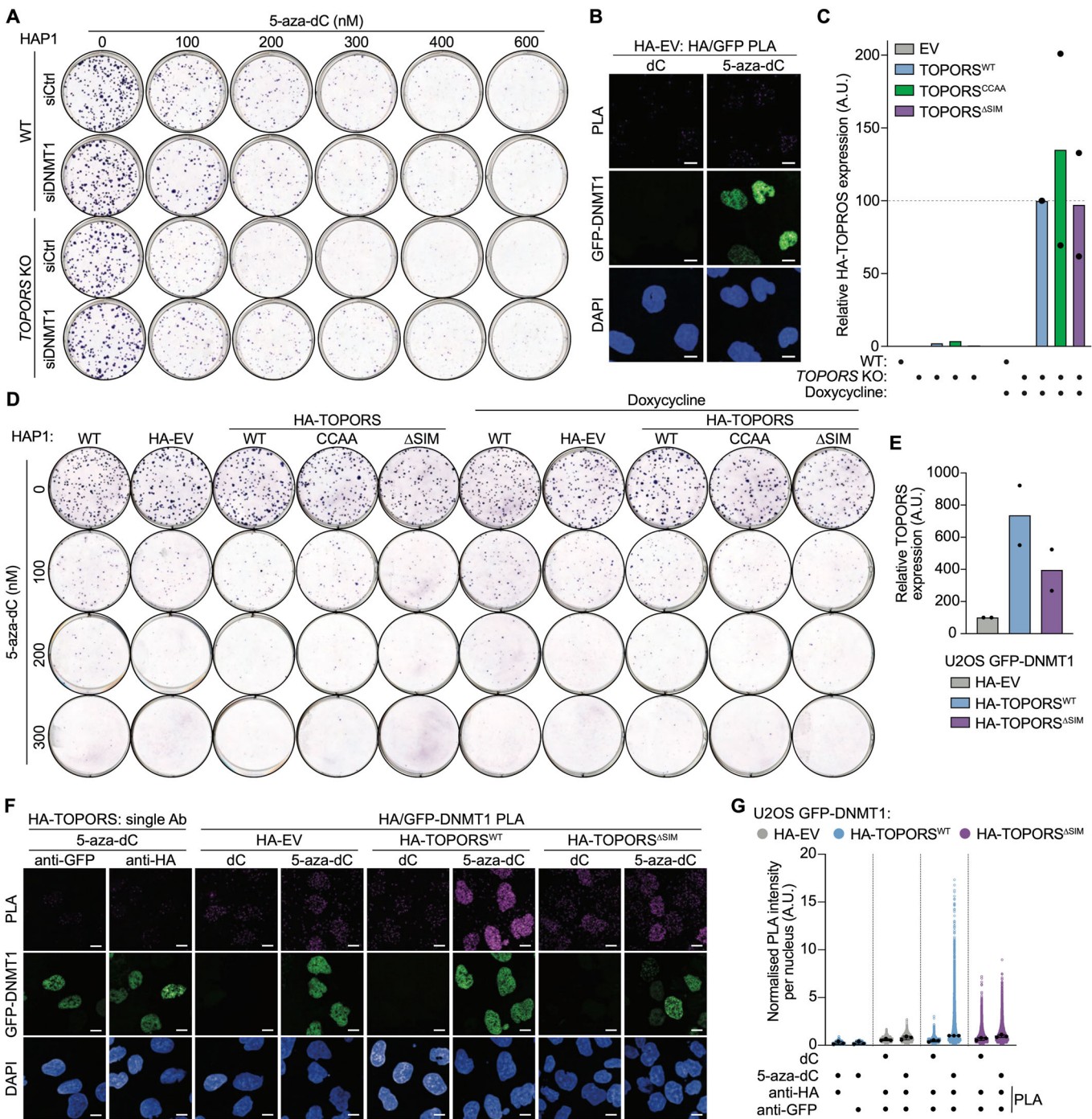

**Figure EV4. TOPORS promotes 5-aza-dC resistance through its RING domain and SUMO-interaction motifs.**

(A) Representative images from clonogenic survival assays in Fig. 4E. (B) Proximity ligation assay in U2OS cells expressing GFP-DNMT1 and HA-EV, treated with dC or 5-aza-dC; quantification in Fig. 4H. (C) Expression levels of HA-TOPORS after doxycycline induction measured by qPCR, relative to GAPDH expression and normalised to doxycycline-induced HA-TOPORS^WT; n = 2 biological replicates performed in technical triplicate, error bars ± SEM. (D) Representative images from clonogenic survival assays in Fig. 5B,C. (E) Expression levels of TOPORS in U2OS GFP-DNMT1 cells measured by qPCR relative to GAPDH and normalised to U2OS GFP-DNMT1 cells expressing HA-EV; n = 2 biological replicates performed in technical triplicate. (F, G) Representative images (F) and quantification (G) of PLA in U2OS GFP-DNMT1 cells expressing HA-EV, HA-TOPORS^WT or HA-TOPORS^ΔSIM treated with dC or 5-aza-dC; scale bars = 10 μm. In (G), black dots display the median normalised PLA intensity of each biological replicate for each condition; n = 3 independent biological replicates, error bars ± SEM. Source data are available online for this figure.

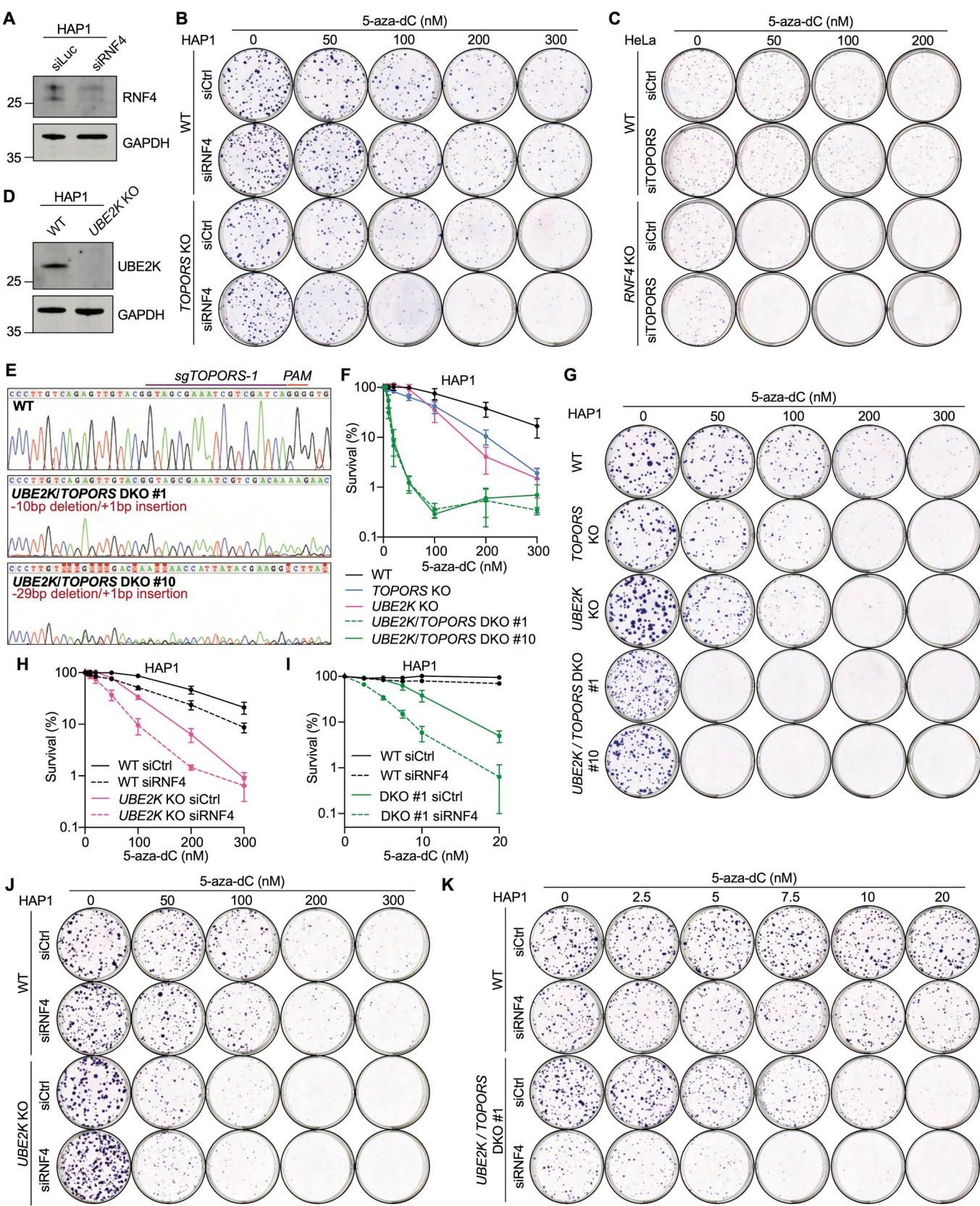

**Figure EV5. TOPORS functions in parallel to RNF4 and UBE2K to promote cellular 5-aza-dC tolerance.**

(A) Western blot of RNF4 from HAP1 cells after siRNA-mediated depletion of RNF4; representative of two independent replicates. (B, C) Representative images at selected 5-aza-dC doses from clonogenic survival assays in Fig. 6B (B) and Fig. 6C (C). (D) Western blot of UBE2K in WT and *UBE2K* KO HAP1 cells; representative of two independent replicates. (E) Validation by Sanger sequencing of *UBE2K/TOPORS* DKO HAP1 clones. (F) Clonogenic survival assays in WT, *UBE2K* KO, *TOPORS* KO and *UBE2K/TOPORS* DKO HAP1 cells treated with 5-aza-dC; $n = 3$ biological replicates, error bars ± SEM. (G) Representative images of selected doses from (F). (H, I) Clonogenic survival assays with 5-aza-dC in WT and *UBE2K* KO (H; $n = 4$ biological replicates. Note that two replicates are shared with data shown in Fig. 6B) or *UBE2K/TOPORS* DKO #1 (I; $n = 3$ biological replicates) cells transfected with siCtrl or siRNF4; error bars ± SEM. (J, K) Representative images at selected 5-aza-dC doses from (H) and (I), respectively. Source data are available online for this figure.

