## [Peer Review File · The EMBO Journal]

Decitabine cytotoxicity is promoted by dCMP deaminase DCTD and mitigated by SUMO-dependent E3 ligase TOPORS

Christopher Carnie, Maximilian Götz, Chloe Palma-Chaundler, Pedro Weickert, Amy Wanders, Almudena Serrano-Benitez, Hao-Yi Li, Vipul Gupta, Samah Awwad, Christian Blum, Matylda Sczaniecka-Clift, Jacqueline Cordes, Guido Zagnoli-Vieira, Giuseppina D'Alessandro, Sean Richards, Nadia Gueorguieva, Simon Lam, Petra Beli, Julian Stingele, and Stephen Jackson

Corresponding author(s): Christopher Carnie (cjc225@cam.ac.uk), Stephen Jackson (spj13@cam.ac.uk), Julian Stingele (stingele@genzentrum.lmu.de)

Review Timeline:

Submission Date:	19th Sep 23
Editorial Decision:	17th Oct 23
Revision Received:	15th Mar 24
Accepted:	16th Apr 24

Editor: Hartmut Vodermaier

Transaction Report:

Dr. Christopher James Carnie
University of Cambridge
Cancer Research UK Cambridge Institute
University of Cambridge Li Ka Shing Centre
Robinson Way
Cambridge, Cambridgeshire CB2 0RE
United Kingdom

17th Oct 2023

Re: EMBOJ-2023-115654

The dCMP deaminase DCTD and the E3 ligase TOPORS are central mediators of decitabine cytotoxicity

Dear Dr. Carnie,

Thank you again for submitting your study on mediators of decitabine cytotoxicity for our consideration. We have now received comments from three expert referees, copied below for your information. As you will see, all reviewers consider the question addressed as well as your findings interesting and potentially important. At the same time, they do raise a number of substantive issues that would need to be satisfactorily clarified before publication. Should you be able to adequately address the key points of the referees, we would be happy to pursue a revised manuscript further for publication.

Since it is our policy to aim for only a single round of major revision, I would in this case encourage you to get back to me with a tentative point-by-point response and revision plan already during the early stages of the revision, so that we could discuss the key revision requirements and how the main concerns may be best addressed. As always, our EMBO Press 'scooping protection' policy would mean that competing work appearing elsewhere during this revision will not affect our final decision, so that an extension of the default three-months revision period would also be an option if necessary.

Detailed information on preparing, formatting and uploading a revised manuscript can be found below and in our Guide to Authors. Thank you again for the opportunity to consider this work for The EMBO Journal, and I look forward to hearing from you in due time.

With best regards,

Hartmut

9) Digital image enhancement is acceptable practice, as long as it accurately represents the original data and conforms to community standards. If a figure has been subjected to significant electronic manipulation, this must be clearly noted in the figure legend and/or the 'Materials and Methods' section. The editors reserve the right to request original versions of figures and the original images that were used to assemble the figure. Finally, we generally encourage uploading of numerical as well as gel/blot image source data; for details see: embopress.org/page/journal/14602075/authorguide#sourcedata

At EMBO Press, we ask authors to provide source data for the main manuscript figures. Our source data coordinator will contact you to discuss which figure panels we would need source data for and will also provide you with helpful tips on how to upload and organize the files.

In the interest of ensuring the conceptual advance provided by the work, we recommend submitting a revision within 3 months (15th Jan 2024). Please discuss the revision progress ahead of this time with the editor if you require more time to complete the revisions. Use the link below to submit your revision:

Link Not Available

Referee #1:

In this manuscript, Carnie et al. investigate the determinants of cytotoxicity associated with 5-aza-dC, also known as decitabine, through CRISPR/Cas9 screens. Decitabine is used in the treatment of certain blood disorders, such as MDS and AML. The therapeutic mechanism of 5-aza-dC is believed to be related to its ability to trap DNA methyltransferases (DNMTs) onto DNA, resulting in DNA hypomethylation as well as the formation of DNA-protein crosslinks (DPCs) containing DNMT1. However, the exact mechanisms underlying its action and patient responses remain unclear.

This study used CRISPR/Cas9 screens to identify genes affecting sensitivity or resistance to 5-aza-dC. It revealed that, in DPC-repair-proficient cells, 5-aza-dC's cytotoxicity results from the conversion of 5-aza-dC to 5-aza-dUMP by DCTD, suggesting that the origin of cytotoxicity is independent of DNA hypomethylation and DNMT1-DPCs.

To gain more insights, the authors performed another CRISPR/Cas9 screen, this time in DCTD KO cells, and found that DPC repair pathways become more prominent determinants of 5-aza-dC sensitivity under this condition. Using a modified iPOND technique (called iPOND-DPC) to identify the proximal proteome of DNMT1-DPCs, the authors identified, among other known DPC repair factors, TOPORS (a SUMO1- and ubiquitin E3 ligase) as a factor that confers protection against DNMT1-DPCs. They propose that TOPORS is recruited to decitabine-induced DNMT1-DPCs in a SUMO-dependent but ubiquitin-independent manner, facilitating the proteolysis of DNMT1-DPCs in parallel to RNF4 and UBE2K.

This is an important study that challenges current assumptions regarding decitabine's mechanisms of action. This study also reveals a novel DPC repair mechanism mediated by TOPORS, highlighting the critical role of SUMO modifications in DPC

repair.

The presented data are of high quality. The proposed model is supported both by genetic evidence on 5-aza-dC sensitivities and by cellular data on TOPORS recruitment to SUMOylated DNMT1. The study could be enhanced by further investigations into the mechanism underlying the SUMO-dependency of TOPORS recruitment. Additionally, whether TOPORS' ubiquitin E3 ligase activity is required for the repair of DNMT1-DPCs is an important question to address to substantiate their model that TOPORS polyubiquitinates DNMT1-DPCs for proteasomal degradation. Comments are listed below for the authors to consider during the revision.

Major points:

1. Does DCTD KO result in increased DPC burden during 5-aza-dC treatment? Fig. 2b appears to show that the DNMT1-DPC level is higher in DCTD KO cells compared to WT, suggesting that DCTD might funnel the 5-aza-dC metabolite away from DNMT-DPC formation in WT cells. However, the authors do not seem to acknowledge this, given that they state "DNMT1-DPC induction in DCTD KO cells was similar to that in WT cells, (Fig 2b)" (Line 152).
2. It would be helpful to highlight the ranking of DNMT1 in Fig. 1b and Fig. 2c. In Fig. 1b, DNMT1 KO would not confer resistance as cytotoxicity in WT cells is independent of DNMT1. In Fig. 2c, on the other hand, DNMT1 KO would be expected to provide some resistance to 5-aza-dC if DCTD KO results in increased DNMT1-DPCs (as mentioned in Point #1).
3. How TOPORS promotes DPC tolerance is not addressed sufficiently. Are the RING domain and SIMs of TOPORS important for DNMT1-DPC repair in Fig. 5g? Rescue experiments with WT or mutant TOPORS would help address this question. Additionally, are the TOPORS SIMs important for the DNMT1 interaction in Fig. 4i,j?
4. Does TOPORS contribute to SUMOylation of DNMT1-DPCs? Mailand's group has reported that there are SUMO E3 ligases for DNMT1-DPCs redundant to PIAS4, but TOPORS was not included in their study (Ref #37).
5. Line 251: "While we detected an interaction between WT GFP-TOPORS and a high molecular weight ubiquitylated interactor, this was not overtly 5-aza-dC-inducible but undetectable in co-immunoprecipitates of GFP-TOPORS CCAA (Fig 4i)." Could this be TOPORS auto-ubiquitination? TOPORS pull-down under denaturing condition could clarify this possibility. The same experiment could also be used to test if the SUMO conjugates in the pull-down is reduced under a denaturing condition. That would support authors' interpretation that the SUMO chains are conjugated to interacting proteins (presumably DNMT1) but not to TOPORS itself.

Minor points:

1. Supplemental Fig. 3a,b: Is there any TOPORS antibody that can validate the lack of TOPORS proteins in TOPORS KO and siTOPORS cells?
2. Line 64: "In addition to DNMT1 DNA-protein crosslink (DPC) induction," requires a prior introduction of DNMT1-DPC formation.

Referee #2:

General summary & opinion of study - significance, questions, and findings

The study by Carnie et al seeks to identify factors that mediate cytotoxicity of decitabine (5-aza-dC), a deoxycytidine analogue used in the treatment of haematological malignancies. Decitabine is a nucleoside analogue, a class of therapy which are all prodrugs and thus require bioactivation inside cells. As a result, these therapies can have multiple active metabolites and typically poly-pharmacologic (and poorly resolved) modes of action. Understanding the molecular underpinnings of how these drugs are metabolised and subsequently exert their cytotoxic effects is critical to improving how these therapies are used in the clinic. Thus, I find the question posed in the study, and the findings presented, to be highly relevant and of great interest.

To identify factors controlling decitabine cytotoxicity, the authors employ a multi-omic approach, encompassing a series of whole genome CRISPR screens in a haploid cell line (HAP1), together with a new proteomic approach (based upon iPOND - immunoprecipitation of nascent DNA) which can define the proximal proteome of DNA protein crosslink (DPC) lesions, which are understood to be a component of how decitabine exerts cytotoxicity. By this way, the authors uncover an alternative metabolic route for decitabine and characterise factors involved in DPC-dependent and independent pathways, including the identification of dCMP deaminase (DCTD) and the E3 ligase TOPORS.

Specific major concerns essential to be addressed to support the conclusions

1. Validation of findings in additional relevant cell models

As written by the authors in the introduction, decitabine is clinically used in myelodysplastic syndromes (MDS), acute myeloid leukaemia (AML) and chronic myelogenous leukaemia (CML). The principal cell model used in the study is HAP1, which is derived from KBM-7, a CML line, however they are quite different in morphology and characteristics (for e.g., they are adherent). Other cell lines include the osteosarcoma line U2-OS and RPE-1 cells. I understand these are great models for the screen and

downstream mechanistic experiments, but given the intent is to identify predictive biomarkers for decitabine treatment, additional more disease relevant models (e.g., AML or more CML lines) to suggest wider applicability of the findings (specifically controlling decitabine cytotoxicity) would be an important addition to the study - to validate these key findings more broadly.

2. Relevant study omitted from the manuscript

The first major finding in the study is the identification of DCTD and DCTPP1 (dCTPase) as decitabine resistance factors in the HAP1 cell line, this is shown in Figure 1 and discussed between lines 121 and 133. There is a study - Requena et al., PMID: 27325794, entitled "The nucleotidohydrolases DCTPP1 and dUTPase are involved in the cellular response to decitabine" - which is very relevant to this section and the discussion of these results. This paper shows recombinant DCTPP1 can hydrolyse 5-aza-dCTP (thus, line 124 can be updated with this reference as this has been shown and so no need to speculate that DCTPP1 can catalyse this reaction). The study also suggests TYMS inhibition and uracil misincorporation can be important components of decitabine mode of action. So the findings in the present study - arrived at in an unbiased whole genome screen - further support this previously published model (see model in the above reference). This also calls into question Figure 1c, comparing existing model of decitabine action and additional model based on screen, as TYMS inhibition and DNA incorporation was already proposed in that study. Would revise. Here you are supporting a pre-existing model with genetic evidence in HAP1 cells.

3. Consideration of dNTP pool perturbations upon decitabine cytotoxicity.

Loss of DCTD or DCTPP1 would be expected to lead to an increase in the dCTP pool. This is supported from pombe studies with DCTD (PMID: 22927644) and knockdown studies in human cells with DCTPP1 (PMID: 27325794) (perhaps there are also other examples out there). It's typically thought that active metabolites of nucleoside analogues compete with their endogenous counterparts (e.g., 5-aza-dCTP vs dCTP), and thus an elevation in the dCTP pool could also be expected to give rise to decitabine resistance. Can the authors integrate this into their current model and test the relevant contribution? dCTP is also a feedback inhibitor of DCK - the enzyme which activates decitabine - can the authors verify that in DCTD KO cells decitabine is being phosphorylated to similar levels? To exclude this possible explanation of resistance.

4. Verification of commercial KO lines absent

The absence of the target protein in the HAP1 DCK, DCTD, and TOPORS cell lines are not shown.

Minor concerns that should be addressed

5. Figure legends for western blots

In the figure legends it is not indicated how many experiments the Western blots are representative of. n=?

6. Comparison to DCK KO?

When stating that DCTD KO causes "profound" decitabine resistance (line 127), it would be informative to compare degree of resistance against DCK KO (which is included in the study but not shown) for reference. As here very little to no decitabine should be activated.

7. Toning down language to reflect models used

The title says "central mediators" and numerous points in manuscript discuss results in "wild-type cells" (e.g., line 31, line 102) which reads with very broad implications, but those results are largely obtained in HAP1 cells. So I think either being more specific, or toning down, these sentences would be good.

Additional non-essential suggestions for improving the study (which will be at the author's/editor's discretion)

In addition to the suggested experiments (comment #1), have the authors considered looking at publicly available pharmacogenomic datasets? Decitabine is included in the Broad (CTRP) and maybe other large-scale drug screening datasets (accessible via DepMap) - does decitabine cell killing in the panels of haematological cell lines included (100+) correlate with basal expression of factors identified in the screens? DCTD, TYMS, TOPORS etc. There are many reasons why a correlation wouldn't exist, but if it does, that could support a broader relevance of the findings beyond the cell models used.

Referee #3:

The authors conducted genetic screens to investigate the cytotoxic effects of 5-aza-dC. Their research revealed that the conversion of 5-aza-dC into 5-aza-dUMP, facilitated by the enzyme DCTD, is responsible for its cytotoxicity. Interestingly, the absence of DCTD can lead to resistance to the chemotherapy drug decitabine, despite these cells showing elevated levels of DPC lesions (DNMT1-DNA crosslinks) induced by decitabine, which is commonly used in the treatment of Myelodysplastic syndromes.

To further explore how elevated levels of DNMT1-DNA crosslinks are repaired, the authors conducted another genetic screening in DCTD-KO cells. Their findings indicate that DNMT1-DNA crosslinks are repaired by the SUMO1/ubiquitin E3 ligase TOPORS, which is responsible for degrading these DNMT1-DNA lesions and thereby promoting cell survival.

The findings of this study hold significant importance. Their investigation into the use of decitabine (5-aza-dC) for the treatment of haematological cancers has unveiled its capacity to induce changes in DNA. Notably, the unexpected discovery of DCTD's role in resistance adds a crucial layer to our understanding of how this drug operates.

These insights provide a deeper understanding of the mechanisms behind the cytotoxic effects of decitabine, highlighting the critical role of nucleotide metabolism in its effectiveness. I found this paper interesting and believe it's worth publishing in EMBO J, but it requires major revisions.

There are two major drawbacks in this manuscript. First, the most important and clinically relevant discovery of this manuscript, which is that 5-aza-dC toxicity is based on a deamination pathway regulated by DCDT, and not by DNMT1-DNA crosslinks and consequent hypomethylation, is superficially investigated. To support this statement, further experiments are needed. If their hypothesis is correct, then DCTD-KO cells should strongly depend on the BER pathway for sensitivity to 5-aza-dC. Therefore, the authors should re-sensitize HAP1-DCDT-KO cells to 5-aza-dC by inactivating the BER pathway or the glycosylase responsible for recognizing and repairing this lesion (e.g., UNG, XRCC1).

Second, the role of TOPORS in DNMT1-DNA proteolysis repair should be further supported. Based on the experiments performed, it's not clear whether the TOPORS repair model of DNMT1-protein lesion is post-replicative or replicative. A more detailed mechanistic insight into TOPORS is necessary. It's also unclear how this specific DPC (DNMT1) is finally degraded after being ubiquitinated/sumoylated by TOPORS. Is it through the proteasome or an SPRTN-dependent process?

In summary, these findings have significant implications for both clinical applications and the understanding of DPC repair. However, the study needs significant improvement.

Major Comments:

1. The major discovery of the manuscript regarding 5-aza-dC cytotoxicity acting through its deamination pathway should be further supported. Consider experiments to demonstrate that DCTD-KO cells depend on the BER pathway for sensitivity to 5-aza-dC.
2. In Figure 2b, the authors should analyze the cell cycle profiles of DCTD-KO and WT cells to determine whether the resistance of DCTD-KO cells is due to differences in cell cycle progression. E.g., the resistance of DCTD-KO cells could be due to suppressed cell cycle progression, and consequent arrest of these cells in the G1-phase.
3. In Figure 3, the authors discuss DNMT1-DPC repair mechanism in the context of the post-replicative pathway. However, they are using iPOND, a classical biochemical method to isolate factors associated with DNA replication forks. Moreover, cell synchronization and subsequent treatment clearly indicate the isolation of replication-related proteins. To clarify this, their iPOND experiments must be analyzed with well-recognized DNA replication factors such as PCNA and DNA polymerases delta or epsilon, and done in the presence of a thymidine chase to determine whether DNA replication fork moves in response to 5-aza-dC treatment. In addition, they have identified several major components for replisome disassembly (p97-UBXD7) and ICL-protein FANCA, all of which are also replication-related protein complexes. Altogether, this indicates that their model of DNMT1 repair is very likely replication-dependent.
4. To clarify whether TOPORS is a replication-dependent or independent DPC-repair protein, the authors should conduct iPOND chase experiments and monitor TOPORS recruitment and progression at nascent and mature chromatin, with appropriate controls (PCNA and histone H3).
5. In Figure 4, use a Ub inhibitor in Fig.4g to confirm their mass-spec data (from Fig 3f) and directly demonstrate that the TOPORS recruitment to DNMT1 PLA foci is only SUMO-dependent. To support their statement, they should also use the TOPORS-SIM-mutated variant to demonstrate whether TOPORS works as a STUbL in this pathway or not.
6. In addition, the survival assays in TOPORS-ko cells should be restored with TOPORS-wt and also investigated with TOPORS-variants defective for E3-ub ligase activity and SUMO binding.
7. In lines 158-169, the Rank plot demonstrating the loss of DPC repair genes in DCTD KO cells being sensitive to 5-aza-dC treatment and the increase in DNMT1-DPCs in response to 5-aza-dC cytotoxicity appears to be contradictory to the earlier statement (lines 151-156) where the authors discussed 5-aza-dC cytotoxicity in a DPC-independent manner. This inconsistency should be addressed or clarified.
8. In lines 473-485, in the relevant section, the authors describe the iPOND assay as a modified version, but upon closer examination, it seems that they have essentially followed the intact protocol from the referenced paper. Their method is simply

iPOND and not POND-DPC.

9. Additionally, aside from presenting the data regarding SUMO_i with and without 5-aza-dC as demonstrated in Figure 4j, it is advisable to incorporate experiments involving mutated SUMO binding motif of TOPORS. This would help determine if TOPORS is indeed recruited to the DNMT1-DPCs sites as a STUbL.

10. Considering the possibility of DNMT1-DPC degradation by the proteasome system, it would be intriguing to observe the accumulation of these DPCs in the presence of a proteasome inhibitor.

Minor comments

1. In lines 55-60, the authors effectively address the observed treatment variability in patients, highlighting the significance of low-dose drug exposure in resistance and relapse. Their insightful analysis aligns with a recurring theme in disease management and treatment strategies seen in the broader literature. As seen with SN-38 treatment, a drug causing DNA damage and forming TOP1cc, there is a significant convergence in anticancer drug therapy outcomes (PMID: 35869071, PMID: 37240063). Notably, these treatments also result in unsuccessful repair processes and genetic instability, contributing to the discussion. I recommend that the authors incorporate references to these papers (PMID: 35869071, PMID: 37240063), as they provide possible explanations for patients' responses to these drugs. This inclusion will enhance the comprehensiveness and relevance of the study.

2. In Figure 4 a-f, Supplementary figure 4 e-g: It would be beneficial for the manuscript if the authors could consider including representative images of clonogenic survival assays in the supplementary section.

3. In line 427-436; A minor typographical error is present while mentioning the cell number, which should be written as 2.5×10^8 . The section defines the concentration of the drugs used in the screens; however, it's not quite clear the rationale for using these concentrations. Notably, if a sensitization study has been conducted to test these drugs on the cell lines used in the study, then authors are encouraged to provide the cytotoxicity data in the supplementary section.

4. Also, regarding reference 72 in line 448, the DrugZ algorithm has been utilized to analyze the sequencing data. Authors are encouraged to detail this section separately and provide information on the pipeline, as the DrugZ algorithm takes data that is pre-processed using several algorithms to reach the format where read counts are utilized for computing enrichment/depletion. Additionally, it would be beneficial to include the sequencing depth analysis used in the screens to understand the overall representation of the reads for each sgRNA sequenced.

5. Authors are encouraged to address the overall status of DNA Damage repair factors obtained in the screen and possible interference in the outcome of the model (Fig Supplement 2 d).

Reviewer comments (reproduced in their entirety, our responses in green)

Referee #1:

In this manuscript, Carnie et al. investigate the determinants of cytotoxicity associated with 5-aza-dC, also known as decitabine, through CRISPR/Cas9 screens. Decitabine is used in the treatment of certain blood disorders, such as MDS and AML. The therapeutic mechanism of 5-aza-dC is believed to be related to its ability to trap DNA methyltransferases (DNMTs) onto DNA, resulting in DNA hypomethylation as well as the formation of DNA-protein crosslinks (DPCs) containing DNMT1. However, the exact mechanisms underlying its action and patient responses remain unclear.

This study used CRISPR/Cas9 screens to identify genes affecting sensitivity or resistance to 5-aza-dC. It revealed that, in DPC-repair-proficient cells, 5-aza-dC's cytotoxicity results from the conversion of 5-aza-dC to 5-aza-dUMP by DCTD, suggesting that the origin of cytotoxicity is independent of DNA hypomethylation and DNMT1-DPCs.

To gain more insights, the authors performed another CRISPR/Cas9 screen, this time in DCTD KO cells, and found that DPC repair pathways become more prominent determinants of 5-aza-dC sensitivity under this condition. Using a modified iPOND technique (called iPOND-DPC) to identify the proximal proteome of DNMT1-DPCs, the authors identified, among other known DPC repair factors, TOPORS (a SUMO1- and ubiquitin E3 ligase) as a factor that confers protection against DNMT1-DPCs. They propose that TOPORS is recruited to decitabine-induced DNMT1-DPCs in a SUMO-dependent but ubiquitin-independent manner, facilitating the proteolysis of DNMT1-DPCs in parallel to RNF4 and UBE2K.

This is an important study that challenges current assumptions regarding decitabine's mechanisms of action. This study also reveals a novel DPC repair mechanism mediated by TOPORS, highlighting the critical role of SUMO modifications in DPC repair.

The presented data are of high quality. The proposed model is supported both by genetic evidence on 5-aza-dC sensitivities and by cellular data on TOPORS recruitment to SUMOylated DNMT1. The study could be enhanced by further investigations into the mechanism underlying the SUMO-dependency of TOPORS recruitment. Additionally, whether TOPORS' ubiquitin E3 ligase activity is required for the repair of DNMT1-DPCs is an important question to address to substantiate their model that TOPORS polyubiquitinates DNMT1-DPCs for proteasomal degradation. Comments are listed below for the authors to consider during the revision.

We thank the reviewer for their supportive comments and helpful feedback.

Major points:

1. Does DCTD KO result in increased DPC burden during 5-aza-dC treatment? Fig. 2b appears to show that the DNMT1-DPC level is higher in DCTD KO cells compared to WT, suggesting that DCTD might funnel the 5-aza-dC metabolite away from DNMT-DPC formation in WT cells. However, the authors do not seem to acknowledge this, given that they state "DNMT1-DPC induction in DCTD KO cells was similar to that in WT cells, (Fig 2b)" (Line 152).

The observed increase in DNMT1-DPCs in *DCTD* KO cells was not apparent in all of our replicates. Overall, *DCTD* KO cells seem to experience slightly less DNMT1-DPC induction than their WT counterparts - we have replaced the original figure with one more representative (new Fig 2B) and also show a further replicate below. However, this modest reduction in DPC formation is unlikely to account for the dramatic resistance phenotype

given that siRNA-mediated depletion of DNMT1 did not confer 5-aza-dC resistance in either WT or *DCTD* KO cells (new Fig EV1D-F). This is in contrast to *TOPORS* KO cells, in which siDNMT1 fully restores 5-aza-dC sensitivity to WT levels (new Fig 4E-F), indicating that 5-aza-dC-induced DNMT1-DPCs contribute to toxicity in cells with defective DPC repair, but not in WT or *DCTD* KO cells.

New Fig 2B

additional replicate of Fig 2B

New Fig EV1E

New Fig EV4E

New Fig 2. (B) DNMT1-DPC formation assessed by PxDNA in WT, *DCTD* KO and *DCK* KO HAP1 cells treated with 5-aza-dC (1 μM) for the indicated times; representative of 3 independent experiments. New Fig EV1. (E) Clonogenic survival assays with siRNA-transfected WT and *DCTD* KO cells from treated with 5-aza-dC; n = 3 biological replicates, error bars ± SEM. New Fig 4. (E) Clonogenic survival assays in WT and *TOPORS* KO HAP1 cells transfected with indicated siRNAs and treated with 5-aza-dC; n = 3 biological replicates, error bars ± SEM.

2. It would be helpful to highlight the ranking of DNMT1 in Fig. 1b and Fig. 2c. In Fig. 1b, DNMT1 KO would not confer resistance as cytotoxicity in WT cells is independent of DNMT1. In Fig. 2c, on the other hand, DNMT1 KO would be expected to provide some resistance to 5-aza-dC if *DCTD* KO results in increased DNMT1-DPCs (as mentioned in Point #1).

We have added DNMT1 to these plots as requested (new Fig 1B and new Fig 2C). DNMT1 does not score as a hit in any of these screens, consistent with our observation described above that siRNA-mediated DNMT1 depletion does not confer 5-aza-dC resistance to either WT or *DCTD* KO cells.

New Fig 1B

New Fig 2D

New Fig 1. (B) Rank plot displaying selected hits from CRISPR/Cas9 screen with 5-aza-dC; dotted lines at NormZ scores of +3/-3 indicate threshold for resistance/sensitivity hits, respectively. New Fig 2. (C) Rank plot of a genome-wide CRISPR/Cas9 screen in *DCTD* KO HAP1 cells treated with 5-aza-dC; dotted lines at NormZ scores of +3/-3 indicate threshold for resistance/sensitivity hits, respectively.

3. How TOPORS promotes DPC tolerance is not addressed sufficiently. Are the RING domain and SIMs of TOPORS important for DNMT1-DPC repair in Fig. 5g? Rescue experiments with WT or mutant TOPORS would help address this question. Additionally, are the TOPORS SIMs important for the DNMT1 interaction in Fig. 4i,j?

As requested, we have performed structure-function analysis of TOPORS in clonogenic survival assays, immunoprecipitation and PLA experiments. Indeed, inducible re-expression of TOPORS^{WT} rescued the 5-aza-dC sensitivity of *TOPORS* KO cells, but this was not the case for expression of TOPORS' RING domain mutants (TOPORS C103A C1036A; TOPORS^{CCAA}) or TOPORS mutated in the 6 candidate SUMO-interaction motifs (SIMs; TOPORS^{ΔSIM}; new Fig 5B-C, new Fig EV4C-E). Furthermore, we established that ubiquitin ligase-defective TOPORS^{CCAA} retains its ability to interact with high molecular weight SUMOylated proteins and DNMT1 in a 5-aza-dC-inducible manner (new Fig 5D), while TOPORS^{ΔSIM} does not (new Fig 5F), suggesting that TOPORS is recruited through its SIMs to SUMOylated DNMT1-DPCs where its ubiquitin ligase activity is important for polyubiquitylation and subsequent degradation of the DNMT1-DPCs. Unfortunately it was extremely challenging to establish *TOPORS* KO cells re-expressing TOPORS, so we have not been able to extend these studies to PxP assays due to time constraints. However, we feel that the clonogenic survival assays with RING-mutant TOPORS alongside the existing understanding of DNMT1-DPC repair and effect of TOPORS depletion on DNMT1-DPC degradation strongly support our model that TOPORS polyubiquitylates DNMT1-DPCs.

New Fig 5

New Fig 5. (B-C) Clonogenic survival assays in *TOPORS* KO cell lines with (B) or without (C) doxycycline-induced expression of the indicated forms of TOPORS, treated with 5-aza-dC; n = 4 biological replicates, error bars \pm SEM. (D) Co-immunoprecipitation of GFP from extracts of HeLa cells expressing GFP (EV), GFP-TOPORS^{WT} or GFP-TOPORS^{CCAA}, released from thymidine block into S-phase and treated with dC or 5-aza-dC for 1 hour, followed by western blotting for indicated proteins; representative of 3 independent experiments. (F) Co-immunoprecipitation and western blotting as in (D) but with cells expressing GFP (EV), GFP-TOPORS^{WT} or GFP-TOPORS^{ΔSIM}; representative of 3 independent experiments.

4. Does TOPORS contribute to SUMOylation of DNMT1-DPCs? Mailand's group has reported that there are SUMO E3 ligases for DNMT1-DPCs redundant to PIAS4, but TOPORS was not included in their study (Ref #37).

This is a pertinent question, given the reported SUMO1 ligase activity of TOPORS. In PxP assays, we observe increased SUMOylation (most dramatically SUMO1 modification) of DNMT1-DPCs in cells depleted of both RNF4 and TOPORS (new Fig 6G), a phenomenon also observable in stringent GFP-DNMT1 IPs in cells depleted of just TOPORS (new Fig 6H). These findings suggest that rather than promoting DNMT1-DPC SUMOylation, TOPORS acts downstream of DNMT-DPC SUMOylation, with TOPORS depletion impairing proteasomal degradation of DNMT1-DPCs and thus stabilising SUMOylated DNMT1-DPC intermediates. Combined with our structure-function analyses outlined above, this strengthens our argument that TOPORS is recruited to SUMOylated DNMT1-DPCs and promotes their polyubiquitylation and degradation.

New Fig 6

New Fig 6. (G) HeLa WT or *RNF4* KO were transfected with the indicated siRNAs, synchronised by double thymidine block and treated with 5-aza-dC (10 μM). DNMT1-DPCs were isolated using PxP and analysed by western blotting using the indicated antibodies; representative of 3 independent experiments. Asterisks (*) indicate SPRTN dependent DNMT1 cleavage fragment. (H) Stringent immunoprecipitation of GFP-DNMT1 and subsequent western blotting from U2OS cells constitutively expressing GFP-DNMT1, transfected with siCtrl or siTOPORS and treated with dC or 5-aza-dC as indicated after release from a thymidine block into S-phase; representative of 3 independent experiments.

5. Line 251: "While we detected an interaction between WT GFP-TOPORS and a high molecular weight ubiquitylated interactor, this was not overtly 5-aza-dC-inducible but undetectable in co-immunoprecipitates of GFP-TOPORS CCAA (Fig 4i)." Could this be TOPORS auto-ubiquitination? TOPORS pull-down under denaturing condition could clarify this possibility. The same experiment could also be used to test if the SUMO conjugates in the pull-down is reduced under a denaturing condition. That would support authors' interpretation that the SUMO chains are conjugated to interacting proteins (presumably DNMT1) but not to TOPORS itself.

As suggested by the reviewer, we have performed stringent GFP-TOPORS^{WT} and GFP-TOPORS^{CCAA} pull-downs in 1M NaCl and it does indeed appear that the ubiquitylated factor is TOPORS itself, with the ubiquitylation signal not appearing in stringent IPs of GFP-TOPORS^{CCAA} (new Fig 5E). A single band of similar size to unmodified GFP-TOPORS does remain, possibly suggesting some residual TOPORS-independent ubiquitylation.

Interestingly, we were also able to detect conjugates of both SUMO2/3 and SUMO1 in these experiments, but these were not appreciably 5-aza-dC-inducible and might therefore account for some of the background SUMO signal seen in dC-treated IP samples in other experiments (such as in new Fig 5D, shown above). These SUMO signals were increased in stringent immunoprecipitates from GFP-TOPORS^{CCAA}, but the reason for this remains unclear. Overall, this experiment suggests that TOPORS is endogenously SUMOylated and autoubiquitylated, but that the 5-aza-dC-inducible portion of heavily SUMOylated interactors likely represents DNMT1.

New Fig 5

New Fig. 5. (E) Western blots following co-immunoprecipitation of the indicated GFP expression constructs from HeLa cells under stringent, denaturing conditions after 5-aza-dC treatment in S-phase-synchronized cells; representative of 3 independent experiments.

Minor points:

1. Fig EV. 3a,b: Is there any TOPORS antibody that can validate the lack of TOPORS proteins in TOPORS KO and siTOPORS cells?

Unfortunately, despite testing a wide range of commercially available TOPORS antibodies, we have been unable to identify one that we can independently validate.

2. Line 64: "In addition to DNMT1 DNA-protein crosslink (DPC) induction," requires a prior introduction of DNMT1-DPC formation.

We have rectified this omission and added this explanation.

Referee #2:

General summary & opinion of study - significance, questions, and findings

The study by Carnie et al seeks to identify factors that mediate cytotoxicity of decitabine (5-aza-dC), a deoxycytidine analogue used in the treatment of haematological malignancies. Decitabine is a nucleoside analogue, a class of therapy which are all prodrugs and thus require bioactivation inside cells. As a result, these therapies can have multiple active metabolites and typically poly-pharmacologic (and poorly resolved) modes of action. Understanding the molecular underpinnings of how these drugs are metabolised and subsequently exert their cytotoxic effects is critical to improving how these therapies are used in the clinic. Thus, I find the question posed in the study, and the findings presented, to be highly relevant and of great interest.

To identify factors controlling decitabine cytotoxicity, the authors employ a multi-omic approach, encompassing a series of whole genome CRISPR screens in a haploid cell line (HAP1), together with a new proteomic approach (based upon iPOND - immunoprecipitation of nascent DNA) which can define the proximal proteome of DNA protein crosslink (DPC)

lesions, which are understood to be a component of how decitabine exerts cytotoxicity. By this way, the authors uncover an alternative metabolic route for decitabine and characterise factors involved in DPC-dependent and independent pathways, including the identification of dCMP deaminase (DCTD) and the E3 ligase TOPORS.

We thank the reviewer for their supportive comments and feedback.

Specific major concerns essential to be addressed to support the conclusions

1. Validation of findings in additional relevant cell models

As written by the authors in the introduction, decitabine is clinically used in myelodysplastic syndromes (MDS), acute myeloid leukaemia (AML) and chronic myelogenous leukaemia (CML). The principal cell model used in the study is HAP1, which is derived from KBM-7, a CML line, however they are quite different in morphology and characteristics (for e.g., they are adherent). Other cell lines include the osteosarcoma line U2-OS and RPE-1 cells. I understand these are great models for the screen and downstream mechanistic experiments, but given the intent is to identify predictive biomarkers for decitabine treatment, additional more disease relevant models (e.g., AML or more CML lines) to suggest wider applicability of the findings (specifically controlling decitabine cytotoxicity) would be an important addition to the study - to validate these key findings more broadly.

We agree with the reviewer that expanding these observations into more clinically-relevant cell lines will be needed to allow extension of our findings to the search for meaningful clinical biomarkers. However, despite efforts to address this point we have been unable to establish efficient gene editing of relevant cell lines as the use of these cell lines is not yet established in our labs.

We have become aware that TOPORS loss has been reported by other groups in conference abstracts to sensitise AML cell lines to 5-aza-dC and we have now cited these abstracts in the manuscript (Truong et al, *Blood* 2022; Kaito et al, *Blood* 2023). Having contacted these groups, we have learned that their work is much more AML-orientated and also at an advanced stage with another journal. We do not want to simply repeat other people's work, so have instead focussed on strengthening the mechanistic understanding of our phenotypic observations.

To reflect our focus on mechanistic studies, we have toned down our claims relating to AML/MDS treatment.

2. Relevant study omitted from the manuscript

The first major finding in the study is the identification of DCTD and DCTPP1 (dCTPase) as decitabine resistance factors in the HAP1 cell line, this is shown in Figure 1 and discussed between lines 121 and 133. There is a study - Requena et al., PMID: 27325794, entitled "The nucleotidohydrolases DCTPP1 and dUTPase are involved in the cellular response to decitabine" - which is very relevant to this section and the discussion of these results. This paper shows recombinant DCTPP1 can hydrolyse 5-aza-dCTP (thus, line 124 can be updated with this reference as this has been shown and so no need to speculate that DCTPP1 can catalyse this reaction). The study also suggests TYMS inhibition and uracil misincorporation can be important components of decitabine mode of action. So the findings in the present study - arrived at in an unbiased whole genome screen - further support this previously published model (see model in the above reference). This also calls into question Figure 1c, comparing existing model of decitabine action and additional model based on screen, as TYMS inhibition and DNA incorporation was already proposed in that study. Would revise. Here you are supporting a pre-existing model with genetic evidence in HAP1 cells.

We thank the reviewer for bringing this relevant and complementary study to our attention. As suggested, we have updated our text, figures and discussion accordingly.

3. Consideration of dNTP pool perturbations upon decitabine cytotoxicity.

Loss of DCTD or DCTPP1 would be expected to lead to an increase in the dCTP pool. This is supported from pombe studies with DCTD (PMID: 22927644) and knockdown studies in human cells with DCTPP1 (PMID: 27325794) (perhaps there are also other examples out there). It's typically thought that active metabolites of nucleoside analogues compete with their endogenous counterparts (e.g., 5-aza-dCTP vs dCTP), and thus an elevation in the dCTP pool could also be expected to give rise to decitabine resistance. Can the authors integrate this into their current model and test the relevant contribution? dCTP is also a feedback inhibitor of DCK - the enzyme which activates decitabine - can the authors verify that in DCTD KO cells decitabine is being phosphorylated to similar levels? To exclude this possible explanation of resistance.

We agree that assessing the impact of DCTD loss on nucleotide metabolism as a whole will likely be important for understanding exactly how DCTD loss confers 5-aza-dC resistance. However, we have been unable to establish this within the timeframe allocated for revisions. Given the competitive nature of this work, we feel that a full mechanistic explanation of DCTD-mediated 5-aza-dC toxicity is beyond the scope of this manuscript.

Importantly, however, elevated dCTP pools or defects in 5-aza-dC phosphorylation (that might be expected in *DCTD* KOs) would be anticipated to cause 5-aza-dC resistance by causing a dramatic reduction in DNMT1-DPC induction. However, our PxP assay in new Fig 2B (see our response to reviewer 1 above) demonstrates at most only a mild reduction in DNMT1-DPCs that is unlikely to account for the dramatic cellular resistance phenotypes. Indeed, DNMT1 depletion by siRNA did not protect WT or *DCTD* KO cells from 5-aza-dC, and only conferred resistance to DPC-repair-deficient *TOPORS* KO cells (see new Fig EV1D-F and Fig 4E-F; see our response to reviewer 1 above).

4. Verification of commercial KO lines absent

The absence of the target protein in the HAP1 DCK, DCTD, and TOPORS cell lines are not shown.

Unfortunately despite testing multiple commercially-available TOPORS antibodies, we are yet to find one that specifically targets TOPORS. With respect to DCTD and DCK, we have now added western blots confirming these knockouts (new Fig EV1A-B).

New Fig EV1

New Fig EV1. (A-B) Western blot in WT and *DCTD* KO (A) or *DCK* KO (B) HAP1 cells with the indicated antibodies; representative of 3 (A) and 2 (B) independent experiments. The DCTD-specific band in (A) is marked with a red asterisk (*).

Minor concerns that should be addressed

5. Figure legends for western blots

In the figure legends it is not indicated how many experiments the Western blots are representative of. n=?

These details have been added throughout.

6. Comparison to DCK KO?

When stating that DCTD KO causes "profound" decitabine resistance (line 127), it would be informative to compare degree of resistance against DCK KO (which is included in the study but not shown) for reference. As here very little to no decitabine should be activated.

We agree that this is an informative comparison and have conducted the recommended clonogenic survival assays, which show that, as expected, *DCK* KO cells are more resistant to 5-aza-dC than *DCTD* KOs (new Fig 1D-E).

New Fig 1

New Fig 1. (D) Clonogenic survival assays in WT and *DCTD* KO HAP1 cells treated with 5-aza-dC and stained 6 days later; n = 3 biological replicates, error bars \pm SEM. (E) Representative images from (D) of cells at selected 5-aza-dC doses.

7. Toning down language to reflect models used

The title says "central mediators" and numerous points in manuscript discuss results in "wild-type cells" (e.g., line 31, line 102) which reads with very broad implications, but those results are largely obtained in HAP1 cells. So I think either being more specific, or toning down, these sentences would be good.

We have adjusted the wording in these places as suggested, including in the title which now reads "The dCMP deaminase *DCTD* and the SUMO-dependent E3 ligase *TOPORS* mediate decitabine cytotoxicity".

Additional non-essential suggestions for improving the study (which will be at the author's/editor's discretion)

In addition to the suggested experiments (comment #1), have the authors considered looking at publicly available pharmacogenomic datasets? Decitabine is included in the Broad (CTRP) and maybe other large-scale drug screening datasets (accessible via DepMap) - does decitabine cell killing in the panels of haematological cell lines included (100+) correlate with basal expression of factors identified in the screens? *DCTD*, *TYMS*, *TOPORS* etc. There are many reasons why a correlation wouldn't exist, but if it does, that could support a broader relevance of the findings beyond the cell models used.

We have checked through public data accessible via DepMap and cBioPortal but have not

identified overt correlations for DCTD, TOPORS or TYMS.

Referee #3:

The authors conducted genetic screens to investigate the cytotoxic effects of 5-aza-dC. Their research revealed that the conversion of 5-aza-dC into 5-aza-dUMP, facilitated by the enzyme DCTD, is responsible for its cytotoxicity. Interestingly, the absence of DCTD can lead to resistance to the chemotherapy drug decitabine, despite these cells showing elevated levels of DPC lesions (DNMT1-DNA crosslinks) induced by decitabine, which is commonly used in the treatment of Myelodysplastic syndromes.

To further explore how elevated levels of DNMT1-DNA crosslinks are repaired, the authors conducted another genetic screening in DCTD-KO cells. Their findings indicate that DNMT1-DNA crosslinks are repaired by the SUMO1/ubiquitin E3 ligase TOPORS, which is responsible for degrading these DNMT1-DNA lesions and thereby promoting cell survival.

The findings of this study hold significant importance. Their investigation into the use of decitabine (5-aza-dC) for the treatment of haematological cancers has unveiled its capacity to induce changes in DNA. Notably, the unexpected discovery of DCTD's role in resistance adds a crucial layer to our understanding of how this drug operates.

These insights provide a deeper understanding of the mechanisms behind the cytotoxic effects of decitabine, highlighting the critical role of nucleotide metabolism in its effectiveness. I found this paper interesting and believe it's worth publishing in EMBO J, but it requires major revisions.

There are two major drawbacks in this manuscript. First, the most important and clinically relevant discovery of this manuscript, which is that 5-aza-dC toxicity is based on a deamination pathway regulated by DCDT, and not by DNMT1-DNA crosslinks and consequent hypomethylation, is superficially investigated. To support this statement, further experiments are needed. If their hypothesis is correct, then DCTD-KO cells should strongly depend on the BER pathway for sensitivity to 5-aza-dC. Therefore, the authors should re-sensitize HAP1-DCDT-KO cells to 5-aza-dC by inactivating the BER pathway or the glycosylase responsible for recognizing and repairing this lesion (e.g., UNG, XRCC1).

Second, the role of TOPORS in DNMT1-DNA proteolysis repair should be further supported. Based on the experiments performed, it's not clear whether the TOPORS repair model of DNMT1-protein lesion is post-replicative or replicative. A more detailed mechanistic insight into TOPORS is necessary. It's also unclear how this specific DPC (DNMT1) is finally degraded after being ubiquitinated/sumoylated by TOPORS. Is it through the proteasome or an SPRTN-dependent process?

In summary, these findings have significant implications for both clinical applications and the understanding of DPC repair. However, the study needs significant improvement.

We thank the reviewer for their supportive comments and feedback, which have helped to improve our study.

Major Comments:

1. The major discovery of the manuscript regarding 5-aza-dC cytotoxicity acting through its deamination pathway should be further supported. Consider experiments to demonstrate that DCTD-KO cells depend on the BER pathway for sensitivity to 5-aza-dC.

We agree with the reviewer that understanding the contribution of BER to 5-aza-dC cytotoxicity is of importance. However, it is a complex question to pick apart, because the BER machinery could feasibly act on genome-embedded 5-aza-dC, 5-aza-dU and dU (dU incorporation upon 5-aza-dC treatment has been noted previously: PMID: 27325794). BER could therefore in principle contribute to 5-aza-dC cytotoxicity in both WT and in *DCTD* KO cells.

Indeed, in our 5-aza-dC screens, XRCC1, a central mediator of SSB repair, appears as a sensitivity hit in both WT and *DCTD* KO cells, supporting the notion that BER and SSB repair are relevant for more than one aspect of decitabine cytotoxicity – we have updated our figures to highlight this hit in our screens (new Fig 1B, new Fig 2D; also see above in our response to reviewer 1). Furthermore, in polyclonal pools of WT and *DCTD* KO cells edited via CRISPR/Cas9 to inactivate PARP1, we have found that both WT and *DCTD* KO cells are sensitised to 5-aza-dC by PARP1 depletion (new Fig EV1H-J). No glycosylases scored as hits in any of our CRISPR screens, and by using genetic and cell biology approaches we have been unable to identify the glycosylase(s) driving BER upon 5-aza-dC treatment, possibly because of redundant effects of different glycosylases on different base lesions (5-aza-dC, 5-aza-dU, dU). We hope to address this question in follow-up work using reconstituted biochemical systems but feel that this lies beyond the scope of the present study.

We have updated our text to incorporate the data mentioned above and to discuss the role of BER in the cellular response to 5-aza-dC more thoroughly.

New Fig EV1

New Fig EV1. (H) Western blot with the indicated antibodies of polyclonal cell populations of WT and *DCTD* KO cells following CRISPR/Cas9-mediated depletion of PARP1 with the indicated sgRNAs; representative of 2 independent experiments. (I) Clonogenic survival assays on cells from (H) treated with 5-aza-dC; n = 3 biological replicates, error bars \pm SEM. (J) Representative images from (I).

2. In Figure 2b, the authors should analyze the cell cycle profiles of *DCTD*-KO and WT cells

to determine whether the resistance of DCTD-KO cells is due to differences in cell cycle progression. E.g., the resistance of DCTD-KO cells could be due to suppressed cell cycle progression, and consequent arrest of these cells in the G1-phase.

As requested, we have performed cell cycle analysis of WT and *DCTD* KO cells, also combining EdU incorporation with 5-aza-dC treatment (new Fig EV1G). The percentage of cells incorporating EdU within the 30-minute pulse was not substantially affected by DCTD loss. Therefore, the resistance of *DCTD* KO cells to 5-aza-dC is not caused by reduced DNA synthesis and 5-aza-dC incorporation.

New Fig EV1

New Fig EV1. (G) Percentage of EdU-positive cells determined by flow cytometry following 5-aza-dC treatment in the indicated cell lines; n = 3 biological replicates, error bars \pm SD.

3. In Figure 3, the authors discuss DNMT1-DPC repair mechanism in the context of the post-replicative pathway. However, they are using iPOND, a classical biochemical method to isolate factors associated with DNA replication forks. Moreover, cell synchronization and subsequent treatment clearly indicate the isolation of replication-related proteins. To clarify this, their iPOND experiments must be analyzed with well-recognized DNA replication factors such as PCNA and DNA polymerases delta or epsilon, and done in the presence of a thymidine chase to determine whether DNA replication fork moves in response to 5-aza-dC treatment. In addition, they have identified several major components for replisome disassembly (p97-UBXDN7) and ICL-protein FANCA, all of which are also replication-related protein complexes. Altogether, this indicates that their model of DNMT1 repair is very likely replication-dependent.

We were probably not clear enough in our introduction as to how 5-aza-dC induces DNMT1-DPCs. These DPCs occur behind the replication fork but in a replication-coupled manner, with DNMT1 and its cofactor UHRF1 interacting with PCNA to methylate newly-synthesised DNA. As such, DNMT1-DPC repair is replication-independent (see Borgermann et al 2019, *EMBO J* and Weickert et al 2023, *Nat Comms*), despite occurring in the vicinity of replisomes. Using iPOND, in combination with 5-azadC labelling, we can isolate postreplicative DNA containing DNMT1-DPCs. However, it is notable that several replication-associated proteins appear as sensitivity hits in our CRISPR screens, which we speculate could arise from remnants of degraded DNMT1-DPCs perturbing replisomes in the next S-phase following initial DPC induction. We have revised our text to make the basis of DNMT1-DPC induction and repair and its relationship with replication clearer.

4. To clarify whether TOPORS is a replication-dependent or independent DPC-repair protein, the authors should conduct iPOND chase experiments and monitor TOPORS recruitment and progression at nascent and mature chromatin, with appropriate controls (PCNA and histone H3).

As stated above, DNMT1-DPC repair has already been shown to be replication-independent. Therefore, we feel that attempting the proposed experiment would not improve the study. Furthermore, we are unfortunately not able to detect TOPORS via iPOND-western blots due to the lack of suitable reagents.

5. In Figure 4, use a Ub inhibitor in Fig.4g to confirm their mass-spec data (from Fig 3f) and directly demonstrate that the TOPORS recruitment to DNMT1 PLA foci is only SUMO-dependent. To support their statement, they should also use the TOPORS-SIM-mutated variant to demonstrate whether TOPORS works as a STUbL in this pathway or not.

We have performed the PLA experiments suggested by the reviewer. In agreement with our iPOND data, in contrast to SUMO E1i, Ub E1i did not compromise TOPORS recruitment to DNMT1-DPCs as assessed by PLA (new Fig 4G-H). HA-TOPORS^{ΔSIM} was expressed less at the RNA level than HA-TOPORS^{WT} (new Fig EV4E) but had higher PLA signal in dC-treated cells than HA-TOPORS^{WT} (new Fig EV4F-G). Despite this, the PLA signal induction of HA-TOPORS^{ΔSIM} with GFP-DNMT1 upon 5-aza-dC treatment was much less dramatic than that between HA-TOPORS^{WT} and GFP-DNMT1 (new Fig EV4F-G). To support these observations, we used co-immunoprecipitation of GFP-TOPORS^{WT} and GFP-TOPORS^{ΔSIM} and found that GFP-TOPORS^{ΔSIM} is deficient in its ability to interact with high molecular weight SUMO conjugates and heavily modified (presumably SUMOylated) DNMT1 after 5-aza-dC treatment (new Fig 5F). Together, these experiments strengthen the interpretation that TOPORS acts in a SUMO-dependent manner.

New Fig 4

New Fig EV4

New Fig EV4

New Fig 5

New Fig 4. (G) Proximity ligation assay in U2OS cells expressing GFP-DNMT1 and HA-TOPORS, released from a single thymidine block and treated with deoxycytidine (dC), 5-aza-dC and/or Ub E1i or SUMO E1i for 1 hour before pre-extraction of non-chromatin-bound proteins and fixation. (H) Quantification of per-nucleus mean PLA intensities from (G), normalised to the median from the 5-aza-dC-treated U2OS GFP-DNMT1/HA-TOPORS condition. Black dots display the median normalised PLA intensity of each biological replicate for each condition; n = 4 independent biological replicates error bars \pm SEM. **New Fig EV4.** (E) Expression levels of TOPORS in U2OS GFP-DNMT1 cells measured by qPCR relative to GAPDH and normalised to U2OS GFP-DNMT1 cells expressing HA-EV; n = 2 biological replicates performed in technical triplicate. (F-G) Representative images (F) and quantification (G) of PLA in U2OS GFP-DNMT1 cells expressing HA-EV, HA-TOPORS^{WT} or HA-TOPORS^{ASIM} treated with dC or 5-aza-dC. **New Fig 5.** (F) Co-immunoprecipitation and western blotting with HeLa cells expressing GFP (EV), GFP-TOPORS^{WT} or GFP-TOPORS^{ASIM} treated with dC or 5-aza-dC after release from a thymidine block into S-phase; representative of 3 independent experiments.

6. In addition, the survival assays in TOPORS-ko cells should be restored with TOPORS-wt and also investigated with TOPORS-variants defective for E3-ub ligase activity and SUMO binding.

We are pleased to now include data demonstrating that doxycycline-inducible expression of WT TOPORS, but not E3 ligase-defective (TOPORS^{CCAA}) or SUMO-binding-defective mutants (TOPORS^{ΔSIM}) restores 5-aza-dC tolerance of TOPORS KO cells (new Fig 5B-C; new Fig EV4C-D; also shown in this letter in response to Reviewer 1's Major Point 3 above).

7. In lines 158-169, the Rank plot demonstrating the loss of DPC repair genes in DCTD KO cells being sensitive to 5-aza-dC treatment and the increase in DNMT1-DPCs in response to 5-aza-dC cytotoxicity appears to be contradictory to the earlier statement (lines 151-156) where the authors discussed 5-aza-dC cytotoxicity in a DPC-independent manner. This inconsistency should be addressed or clarified.

We recognise that we did not explain our reasoning with sufficient clarity. Having found that DCTD KO cells do not experience substantially fewer DNMT1-DPCs than WT cells upon 5-aza-dC, we concluded that DCTD was driving a mechanism of toxicity independent of DNMT1-DPCs, likely through its deamination of 5-aza-dCMP to 5-aza-dUMP. Consistently, depletion of DNMT1 did not affect 5-aza-dC sensitivity of WT or DCTD KO cells (new Fig EV1D-F, and see above in response to Reviewer 1's Major Point 1). However, if DPC repair factors are lost (e.g. by TOPORS KO), 5-aza-dC-induced DNMT1-DPCs cannot be repaired and thus contribute to toxicity. This is even more apparent in DCTD KO cells, in which the balance between DPC-dependent and DPC-independent cytotoxicity is shifted, resulting in stronger phenotypes upon loss of DPC repair factors upon 5-aza-dC-treatment than in WT cells.

We have updated the text to make this line of reasoning clearer.

8. In lines 473-485, in the relevant section, the authors describe the iPOND assay as a modified version, but upon closer examination, it seems that they have essentially followed the intact protocol from the referenced paper. Their method is simply iPOND and not POND-DPC.

We combined iPOND with 5-azadC co-labelling to identify the proximal proteome of DNMT1-DPCs. Following the reviewer's suggestion, we have rephrased our discussion and now simply refer to these assays as iPOND.

9. Additionally, aside from presenting the data regarding SUMOi with and without 5-aza-dC as demonstrated in Figure 4j, it is advisable to incorporate experiments involving mutated SUMO binding motif of TOPORS. This would help determine if TOPORS is indeed recruited to the DNMT1-DPCs sites as a STUbL.

As suggested, we have mutated the six TOPORS SIMs and performed GFP-TOPORS^{ΔSIM} immunoprecipitations (new Fig 5E; also shown in this letter in response to Reviewer 1, Major Point 5 above). Consistent with TOPORS acting in a SUMO-dependent manner, GFP-TOPORS^{ΔSIM} retained some interaction with unmodified DNMT1 but failed to interact with high molecular weight SUMO conjugates or heavily-modified (presumably SUMOylated) DNMT1.

10. Considering the possibility of DNMT1-DPC degradation by the proteasome system, it would be intriguing to observe the accumulation of these DPCs in the presence of a proteasome inhibitor.

Indeed, DNMT1-DPC repair has already been shown to be entirely dependent on both SUMOylation and proteolysis by the proteasome and SPRTN [see Borgermann et al 2019, Liu et al 2021 and Weickert et al 2023]. We have revised our text to make this point clearer.

Minor comments

1. In lines 55-60, the authors effectively address the observed treatment variability in patients, highlighting the significance of low-dose drug exposure in resistance and relapse. Their insightful analysis aligns with a recurring theme in disease management and treatment strategies seen in the broader literature. As seen with SN-38 treatment, a drug causing DNA damage and forming TOP1cc, there is a significant convergence in anticancer drug therapy outcomes (PMID: 35869071, PMID: 37240063). Notably, these treatments also result in unsuccessful repair processes and genetic instability, contributing to the discussion. I recommend that the authors incorporate references to these papers (PMID: 35869071, PMID: 37240063), as they provide possible explanations for patients' responses to these drugs. This inclusion will enhance the comprehensiveness and relevance of the study.

We have now included the suggested discussion and cited the recommended papers.

2. In Figure 4 a-f, Supplementary figure 4 e-g: It would be beneficial for the manuscript if the authors could consider including representative images of clonogenic survival assays in the supplementary section.

As suggested, we have now included representative images for all clonogenic survival assays shown in the manuscript.

3. In line 427-436; A minor typographical error is present while mentioning the cell number, which should be written as 2.5×10^8 . The section defines the concentration of the drugs used in the screens; however, it's not quite clear the rationale for using these concentrations. Notably, if a sensitization study has been conducted to test these drugs on the cell lines used in the study, then authors are encouraged to provide the cytotoxicity data in the supplementary section.

We have corrected the error and thank the reviewer for spotting it. Regarding IC values, we will add details on how they were determined but do not feel that showing toxicity effects within the screen will be informative. In screens in the Jackson lab, we almost always see slightly different eventual toxicities throughout the screen from the pre-calculated values, presumably in large part due to the effects of the gRNA library and knockouts themselves influencing toxicity at the population level.

4. Also, regarding reference 72 in line 448, the DrugZ algorithm has been utilized to analyze the sequencing data. Authors are encouraged to detail this section separately and provide information on the pipeline, as the DrugZ algorithm takes data that is pre-processed using several algorithms to reach the format where read counts are utilized for computing enrichment/depletion. Additionally, it would be beneficial to include the sequencing depth analysis used in the screens to understand the overall representation of the reads for each sgRNA sequenced.

As requested, we have included further details of the NGS and analysis used in CRISPR screens as a new paragraph. In terms of pre-processing prior to DrugZ, we have provided a link through which readers can access custom scripts used to quantify reads and map them to the Brunello library. The screens were performed with a median mapped read count of 466 ± 152 (mean \pm SD); we have updated our methods section to include this.

5. Authors are encouraged to address the overall status of DNA Damage repair factors obtained in the screen and possible interference in the outcome of the model (Fig Supplement 2 d).

We have added a comment on the other DDR pathways represented by hits in the iPOND experiments summarised in Supp Fig 2D in the relevant part of the results section.

Dr. Christopher James Carnie
University of Cambridge
Cancer Research UK Cambridge Institute
University of Cambridge Li Ka Shing Centre
Robinson Way
Cambridge, Cambridgeshire CB2 0RE
United Kingdom

16th Apr 2024

Re: EMBOJ-2023-115654R
Decitabine cytotoxicity is promoted by dCMP deaminase DCTD and mitigated by SUMO-dependent E3 ligase TOPORS

Dear Dr. Carnie,

Thank you for submitting your final revised manuscript for our consideration. I am pleased to inform you that in light of the positive re-reviews copied below, we have now accepted it for publication in The EMBO Journal.

Yours sincerely,

Hartmut Vodermaier

Referee #1:

In the revised manuscript, Carnie et al. have addressed all my concerns. The newly included data using the RING and SIM mutants of TOPORS strengthened their conclusion that TOPORS is recruited to SUMOylated DNMT1-DPCs and promotes their degradation, thereby protecting cells from 5-aza-dC cytotoxicity. Below, I have listed one more point for the authors to consider, but otherwise, I believe that the manuscript is now suitable for publication in EMBO J.

Major point:

1. It is recommended to reevaluate the clarity of the title. "... DCTD and ... TOPORS mediate decitabine cytotoxicity" might sound as if TOPORS enhances decitabine cytotoxicity similar to DCTD, despite its actual role being the opposite (protecting from the cytotoxicity).

Referee #2:

The authors have addressed my major concerns, either by providing new data or toning down writing in the manuscript. Congrats to the authors on an important study.
